# A Collection of Wet Beam Models for Wave-Ice Interaction

Sasan Tavakoli[1,2] and Alexander V. Babanin[1]

[1]Department of Infrastructure Engineering, The University of Melbourne, Parkville, 3051, VIC, Australia
[2]Department of Mechanical Engineering, Aalto University, Espoo, Finland

**Correspondence:** Sasan Tavakoli (sasan.tavakoli@aalto.fi)

**Abstract.** Theoretical models for the prediction of decay rate and dispersion process of gravity waves traveling into an integrated ice cover expanded over a long way are introduced. The term "*wet beam*" is chosen to refer to these models as they are developed by incorporating water-based damping and added mass forces. Presented wet beam models differ from each other according to the rheological behavior considered for the ice cover. Two-parameter viscoelastic solid models accommodating Kelvin-Voigt (KV) and Maxwell mechanisms along with a one-parameter elastic solid model are used to describe the rheological behavior of the ice layer. Quantitative comparison between the landfast ice field data and model predictions suggests that wet beam models, adopted with both KV and Maxwell mechanisms, predict the decay rate more accurately compared to a dry beam model. Furthermore, the wet beam models, adopted with both KV and Maxwell mechanisms, are found to construct decay rates of disintegrated ice fields, though they are built for a continuous ice field. Finally, it is found that wet beam models can accurately construct decay rate curves of freshwater ice, though they cannot predict the dispersion process of waves accurately. To overcome this limitation, three-parameter solid models, termed Standard Linear Solid (SLS) mechanisms, are suggested to be used to re-formulate the dispersion relationship of wet beam models, which were seen to construct decay rates and dispersion curves of freshwater ice with an acceptable level of accuracy. Overall, the two-parameter wet beam dispersion relationships presented in this research are observed to predict decay rates and dispersion process of waves traveling into actual ice covers, though three-parameter wet beam models were seen to reconstruct the those of freshwater ice formed in a wave flume. The wet beam models presented in this research can be implemented in spectral models on a large geophysical scale.

## 1   Introduction

Mutual interaction between water waves and ice is a multi-physical problem, frequently occurring in polar seas where waves can penetrate ice covers, traveling over kilometers until they die out. The phase and group speeds of the resulting gravity-flexural waves advancing through the ice can be different from that of an open-water sea owing to the effects of forces caused by solid motions. There is a pressing need to understand the mutual effects of ice and gravity waves on each other due to the recent retreat of the sea ice in the Arctic (Stroeve et al., 2008; Comiso et al., 2008; Meier et al., 2013), and the emergence of large and powerful wind-generated water waves in the Antarctic (Young et al., 2011), which can affect the ice-extent (Kohout et al., 2014) by breaking the ice. Specifically, the beak-up of ice may potentially lead to evolution of larger waves. Hence, it is very important to understand the pattern of wave propagation in polar seas on larger geophysical scales. Wave modelers aiming to numerically simulate wave propagation in polar regions need to use proper formulations to calculate the ice-induced energy

damping and the group speed of waves traveling into ice covers (Rogers and Orzech, 2013; Liu et al., 2020). These two can be helpful in the prediction of the amount of wave energy traveling into an ice cover, which can be calculated through spectral modeling of waves on a large geophysical scale, and if they are combined with sea ice break-up models, they can also provide us with evolution of Marginal Ice Zones (e.g. in Kousal et al. (2022)).

Mathematical modeling of the wave-ice interaction has firstly received the attentions of researchers in the 19th century. The first model was developed by Greenhill (1886), who formulated harmonic linear motions in a fluid domain covered with an elastic beam. To build the model, he assumed that the ice extent was spanned over an infinite way and the solid body had relatively small motions. This model lacks energy damping, but, instead, it can be employed in the prediction of the dispersion processes, the result of which is observed to be consistent with the physics of long integrated bodies covering water.

The Greenhill study is the kernel of the next generation of models established for the prediction of mutual effects of water waves and ice. Researchers looking into the wave-ice interaction developed models by modifying the original model of Greenhill. Early developments of models dates back to years between the 1950s and 1970s when scientists had become able to reach polar seas, recording the wave climate. The energy of waves traveling through the ice was observed to be reduced by sea ice (Robin, 1963; Wadhams, 1972) and the phase speed was observed to be affected by the ice. The Greenhill model lacked the former since it was developed for an elastic solid body.

Following the Greenhill's model, various linear mathematical models of wave-ice interaction have been developed. In some of them, interactions between water waves and finite length ice floes have been modeled. Using such an assumption, scattering and radiation problems can be addressed, and motions of a flexible ice floe can be found (e.g. Meylan and Squire (1996); Smith and Meylan (2011); Meylan et al. (2020)). Such a problem has been solved in frequency or time domains (e.g. Meylan and Sturova (2009); Hartmann et al. (2022)), and also shares similarities with engineering problems within field of marine hydroelasticty (e.g. ship hydroelasticity and seakeeping of very large floating structures, e.g. in Hirdaris et al. (2003)).

Some other models highlight mutual interaction of water waves with an infinite length ice cover. Simply stated, ice cover is viewed as a continuum medium. When this approach is embarked, the ice-induced energy decay is found through finding the root of the dispersion equation. To address the energy decay rates of continuum models, scholars have mostly prescribed viscoelastic (*e.g.* Squire and Allan (1977); Wang and Shen (2010)) or proelastic (Chen et al. (2019); Xu and Guyenne (2022)) behavior for the ice, or presumed that a thin layer of viscous fluid (*e.g.* De Carolis and Desiderio (2002)) can be representative of the ice behavior. In addition, in some other models, a damping term, which represents a fluid based dissipation mechanism is considered, though the ice is assumed to be elastic. Such an approach has been introduced by Robinson and Palmer (1990), and is known as RP model. Researchers employing this approach have prescribed elastic behaviour for the ice cover (Squire et al. (2009); Williams et al. (2013)). This fluid damping term has also been combined with scattering models (Williams et al. (2017)). To consider the scattering of water waves by an infinite length cover, or wave reflection by cracks or variation in the thickness, matching methods have been used (e.g. Fox and Squire (1991); Kohout and Meylan (2006); Kohout et al. (2007); Zhao and Shen (2013)).

Models developed for the prediction of decay rates and the dispersion process of continuously integrated ice have been mostly developed by assuming small displacement for the cover. Thus, they solve the solid dynamic problem by using an

Euler-Bernoulli beam theory. Their applications have been seen to be limited. This can be due to the reason that they have mostly developed by simplifying the problem, neglecting some aspects of the fluid and solid motions. Studies performed in the recent decade provide a clear picture of this fact. In some studies concerned with the dispersion process of waves advancing in ice (or elastic) covers, formulated dispersion relationships were reported to reconstruct the dispersion plot with an effective value of rigidity (or Young Modulus), which is much smaller than what was measured in dry tests (Langleben, 1962; Sakai and Hanai, 2002; Cheng et al., 2019). In some other studies, different values of ice viscosity were seen to give the best fitting for energy damping of various frequencies, which may not agree with reality (Marchenko et al. (2021)).

To overcome the limitations of available continuum models, the role of different mechanisms in energy dissipation and the parameters influencing the dispersion process of waves propagating in ice should be understood well. Reviewing the structure of all developed models (example of review papers: Squire et al. (1995); Squire (2007); Zhao et al. (2015); Collins et al. (2016); Squire (2020)), one can conclude that energy dissipation is either assumed to be triggered by solid motions (*e.g.* Wang and Shen (2010)) or by fluid motion, such as fluid damping, overwash or shear stresses (*e.g.* Liu and Mollo-Christensen (1988); Mosig et al. (2015); Herman et al. (2019); Huang et al. (2019)). Furthermore, the dispersion process is dependent on ice flexural rigidity, thickness and its density in most of the models, though added mass can also affect the dispersion process. New continuum models can still be developed by assuming that the fluid-based and solid-based energy damping mechanisms emerge at the same time. The coexistence of solid-based and fluid-based dissipation mechanisms in formulations may help us predict the decay rates with fewer limitations and solve the fluid-solid problem for a more realistic condition. This means that, unlike the previous studies which presented dispersion relationship based on the RP model by assuming that ice is an elastic material, viscoelastic behavior of the ice can also be considered. This provides us with more options in building the dispersion relationship as viscoelastic behavior of the ice cover can also be formulated using different models (e.g. Kelvin-Voigt and Maxwell), which may allow for incorporation of viscosity in flexural rigidity.

The present paper aims to develop continuum wave-ice interaction models by hypothesizing that fluid forces and solid forces emerge simultaneously. Distinguishing the developed models from the previous studies, fluid forces, including damping and added mass, are hypothesized to emerge under a viscoelastic ice beam covering water, vibrating due to the wave forces. To understand the role of rheological behaviour of ice, two different two-parameter solid mechanisms, linking stress and strains arising in the solid beam, are employed to establish these models. Accordingly, two new viscoelastic models with incoporation of fluid damping force are introduced. It is also aimed to provide a better understanding of how different rhological patterns can be implemented in the prediction of decay rate of the ice-covered water and seas. This paper is structured as follows. Models are developed in Section 2. In Section 3, results including wave height decay rates and dispersion curves are presented. At the first step, sensitivities of models to different parameters, shear modulus and viscosity, are analyzed. At the next step, predictions of models are compared against thoese of field and flume measurements. Afterward, two three-parameter solid models are also used to formulate dispersion relationships to investigate the way consideration of more complex rheological behaviour can affect the results. Finally, a discussion on the ability of models in the prediction of decay rates and dispersion curves is presented. In Section 4, concluding remarks and suggestions are presented. Appendix A presents an example of dimensional data.

## 2 Models

### 2.1 Development of Models

Consider a two-dimensional fluid domain containing water. The domain extent is stretched over an infinite length and has a depth of $D$. An ice sheet covers the upper surface of the domain. It implies that no air-water interaction occurs at all. A schematic of the domain is shown in Figure 1 (panel c). Water waves propagate in this domain and their energy is dissipated over time. Assuming that wave height is decayed exponentially, wave height at a longitudinal position of $x$ is formulated as

$$H(x) = H_0(x)e^{-\alpha(x-x_0)}. \tag{1}$$

In Equation 1, $\alpha$ is the decay rate of wave height, and $H_0(x)$ is the wave height at $x = x_0$. In addition, as was explained in Section 1, the wavelength of ice-covered sea (the distance between two consecutive wave crests) can be different from that of open-water condition (see Figure 1 b). Scattering is not considered in the problem, as the cover to have an infinite length.

Let the fluid to be ideal and irrotational. Hence, the fluid motion is represented by the potential field, which is indicated with $\Phi(\mathbf{x}, t)$. Assuming that fluid has a linear cyclic motion with a frequency of $\omega$, the potential field can be re-written in the frequency domain, as per

$$\Phi(\mathbf{x}; \mathrm{t}) = \mathrm{Re}\left[\phi(\mathbf{x})e^{-i\omega t}\right]. \tag{2}$$

The Laplace Equation holds the fluid domain:

$$\nabla^2 \phi(\mathbf{x}) = 0 \qquad\qquad -\infty < x < \infty \qquad\qquad -D < z < \xi. \tag{3}$$

Here, $\xi$ is the elevation of the upper layer of the fluid domain with respect to the calm water line.

The vertical component of velocity is zero at the sea-bed, which signifies that

$$\partial_z \phi(\mathbf{x}) = 0 \qquad\qquad -\infty < x < \infty \qquad\qquad z = -D. \tag{4}$$

The solid body covering the upper layer of the water is assumed to be very thin and its thickness is denoted with $h$. Assuming linearity, vertical motion of the upper layer is expressed as $\mathrm{Re}\left[\xi(\mathbf{x})e^{-i\omega t}\right]$. In the absence of fluid forces, the solid layer follows the Euler-Bernoulli beam theory, as per

$$-\omega^2 \left(\rho_{\mathrm{i}} h\right)\xi + \frac{G_E h^3}{6\left(1-\nu\right)}\xi_{xxxx} = p \qquad\qquad -\infty < x < \infty, \tag{5}$$

where $G_E$ is the dynamic shear modulus of the material, which shall be introduced later. $p$ is the pressure acting on the beam, causing the vibration. Also,

It is assumed that fluid forces emerge when the beam interacts with the water, which are given as

$$f = \omega^2 A\xi + i\omega B\xi. \tag{6}$$

Here, $A$ is the added mass coefficient and $B$ is damping force per unite area. The damping term was previously used for formulation of dispersion relationship of a pure elastic beam by Robinson and Palmer (1990). These two coefficients are phenomenological. Equation 5 can be extended to

$$\left(\rho_w g - \omega^2\left(\rho_i h + A\right)\right)\xi - i(\omega B)\xi + \frac{G_E h^3}{6\left(1-\nu\right)}\xi_{xxxx} = p \qquad\qquad -\infty < x < \infty. \qquad (7)$$

When the ice cover is broken, the body-body interaction may emerge, causing an extra dissipation mechanism, which can be implemented through consideration of an artificial damping term (Lu et al. (2010)). But, such a damping mechanism is not considered in the present research as the ice cover is assumed to be integrated.

Equation 7 is an extended version of the Euler-Bernoulli beam model which is adopted for a beam interacting with water. Hence, the term "*wet beam*" is used with an aim to distinguish the model developed based on Equation 7 from what was presented in Equation 5. Time derivative of surface elevation is approximately $\partial_z\phi$, and time derivative of pressure under the beam is approximately $-\rho_w g(\partial_z\phi) + \rho_w\omega^2\phi$. Thus, the boundary condition on the fluid-solid interface is formulated as

$$\left(\rho_w g - \omega^2\left(\rho_i h + A\right)\right)(\partial_z\phi) - i(\omega B)(\partial_z\phi) + \frac{G_E h^3}{6\left(1-\nu\right)}\partial_{xxxx}(\partial_z\phi) = \rho_w\omega^2\phi \qquad\qquad -\infty < x < \infty. \qquad (8)$$

Using the separation of variable, the solution of Laplace Equation can be established as the sum of $e^{ikx}\cosh kz$. Therefore, the general form of the dispersion equation can be established as

$$k\tan(kD) = \frac{\rho_w\omega^2}{\frac{G_E h^3}{6(1-\nu)}k^4 + \rho_w g - \omega^2\left(\rho_i h + A\right) - i(\omega B)}. \qquad (9)$$

As was mentioned, $G_E$ is the dynamic shear stress modulus of the material having cyclic motions which depends on the mechanical behavior of the substance. It can be a complex number. Its real component is storage modulus and its imaginary component is the loss modulus. As explained before, solid ice cover can be either assumed to be elastic or viscoelastic. As such, models have been developed for both elastic and viscolelastic ice covers. As displayed in Figure 1d, Mechanical behaviour of these materials can be shown by using a spring element (which demonstrates the elastic nature of the material) and a dashpot element (which demonstrates the viscous nature of the material).

For an elastic solid body, $G_E$ (dynamic modulus) is given by

$$G_E = G. \qquad (10)$$

where $G$ is the shear modulus of the material. Loss modulus of an elastic material is zero (there is no imaginary component).

For a viscoelastic material, two different two-parameter models can be used. The first one is Kelvin-Voigt (KV) and the second one is Maxwell. For the former, displacements of both elements (i.e. spring and dashpot) are similar, but the resulting forces are different. For a Maxwell material, however, forces emerging in elements are similar, and displacements are different.

For a KV material, the dynamic shear modulus can be written as

$$G_E = G - i\omega\eta. \qquad (11)$$

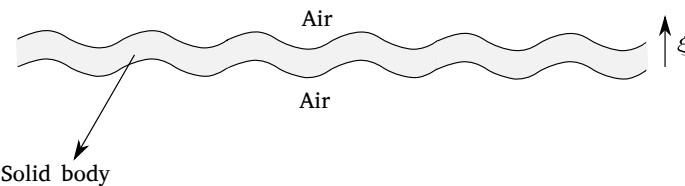

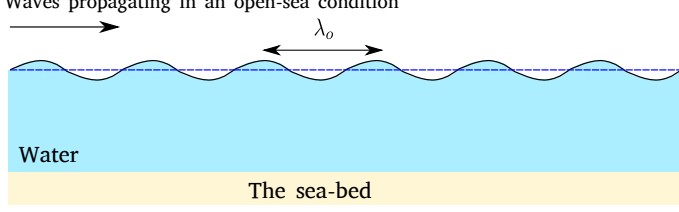

(c) Ice-covered sea

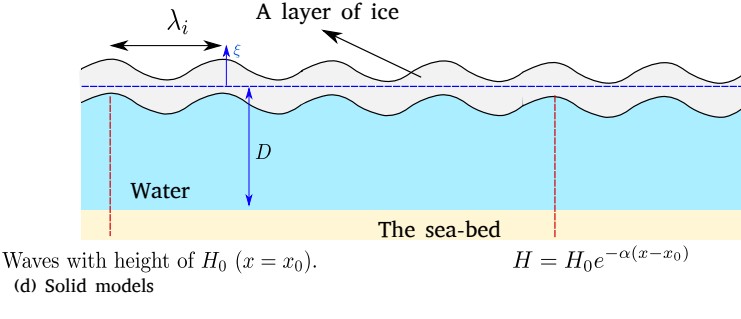

Waves with height of $H_0$ $(x = x_0)$.
(d) Solid models

**Figure 1.** A pictograph of the problem and the different patterns of rheological behavior considered for the ice layer. Panel a and b repetitively show a dry beam and waves propagation in an open-water sea. Panel c shows waves propagating in an ice-covered sea. Panel d shows different rhelogical behavior that can be considered for the ice.

Here, $\eta$ is the dynamic viscosity of the material (Serra-Aguila et al. (2019)). The real and imaginary components of $G_E$ are the storage and loss moduli, as explained before. The former is responsible for harmonic response, and the latter is is responsible for energy dissipation.

For a Maxwell material, the dynamic shear modulus is given by

$$G_E = \frac{G\left(\tau^2\omega^2\right)}{\left(1+\tau^2\omega^2\right)} - i\frac{G(\tau\omega)}{\left(1+\tau^2\omega^2\right)}. \tag{12}$$

where $\tau = \eta/G$ is the relaxation time (Serra-Aguila et al. (2019)). It is worth noting that dynamic viscosity can affect the storage Modulus of Maxwell materials, and shear Modulus can affect the loss Modulus of Maxwell materials.

Dispersion relationships can be established for different materials. The first dispersion relationships is developed for water waves travelling into a pure elastic material, as

$$\omega^2 = \left(\frac{Gh^3}{6\left(1-\nu\right)\rho_w}k^4 + g - i\omega\frac{B}{\rho_w} - \frac{\rho_i(h+A)}{\rho_w}\omega^2\right)k\tanh(kD). \tag{13}$$

Another version of this relationship has been previously documented by Mosig et al. (2015). The only difference between Equation 13 and the one presented in Mosig et al. (2015) is the added mass term, which may slightly increase the wavenumber, which shall be discussed later. This model is called RP. Note that the RP model (dispersion relationship of pure elastic material with zero added mass) was presented in some other studies (e.g., Squire et al. (2009) and Williams et al. (2013)). But note that this model only considers one dissipative mechanism (fluid damping), though in the present research, it is aimed to show that coexistence of two different dissipative mechanism (fluid-based and solid-based) help us predict decay rate and dispersion relationship. The rest of dispersion relationship formulated in the present research incorporates fluid-based and solid-based mechanisms One of the reasons that the pure elastic model is presented in this research is to show how the results of viscoelastic models may differ from those of this model.

The second dispersion relationship describes the link between frequency and wavenumber for a KV material, which is found to be

$$\omega^2 = \left(\frac{Gh^3}{6(1-\nu)\rho_w}k^4 - i\frac{\omega\eta h^3}{6(1-\nu)\rho_w}k^4 + g - i\omega\frac{B}{\rho_w} - \frac{\rho_i(h+A)}{\rho_w}\omega^2\right)k\tanh(kD). \tag{14}$$

Equation 14 with $B = 0$ is known as FS model (Mosig et al. (2015)). Dispersion relationship of a Maxwell model is built as

$$\omega^2 = \left(\frac{G\left(\tau^2\omega^2\right)h^3}{6(1-\nu)\left(1+\tau^2\omega^2\right)\rho_w}k^4 - i\frac{G(\tau\omega)h^3}{6(1-\nu)\left(1+\tau^2\omega^2\right)\rho_w}k^4 + g - i\omega\frac{B}{\rho_w} - \frac{\rho_i(h+A)}{\rho_w}\omega^2\right)k\tanh(kD). \tag{15}$$

Each of the above equations provides us with the roots of the dispersion relationships. We choose the dominant root with the positive real part, which refers to the wave advancing in the solid body. The dominant root is

$$k = k_i - i\alpha, \tag{16}$$

and is found through using a numerical method with an initial guess. All the dispersion relationships have multiple roots. We seek for the one found with an initial guess which is set to be very close to wavenumber of the open-water condition. In Equation 16, $k_i$ is the wavenumber in the ice-covered sea and $\alpha$ is the wave height decay rate, which were introduced before. If the initial guess is set to be much greater than the open-warer wavenumber, abrupt changes in the $\alpha$ vs $\lambda_o$ may emerge. Readers who are interested in finding the root of the dispersion relationship are referred to Zhao et al. (2017) and Das (2022).

## 2.2 Non-dimensional representation

To analyze results and to use any of the models more easily, all parameters are normalized. Parameters are normalized using the Buckingham Pi-theorem, enabling us to perform scaling law. For the first dispersion relationship, eight parameters are involved, but for the second and third relationships, nine parameters are involved. Therefore, five non-dimensional numbers are identified for the pure elastic model, and six non-dimensional numbers are identified for models developed for viscoelastic materials.

Non-dimensional numbers have been previously presented by different authors, concerted with the field wave-mud or wave-ice interaction (Jain and Mehta, 2009; Yu et al., 2019, 2022). The first non-dimensional number represents the non-dimensional wavelength in an open-water condition, which is given by

$$\hat{\lambda}_o = \lambda_o/h. \tag{17}$$

The second non-dimensional number is the non-dimensional wavenumber of waves propagating in a covered sea condition, given by

$$\hat{k}_i = k_i/k_o. \tag{18}$$

Here, $k_o$ is the open-water wave number. The attenuation rate is normalized as

$$\hat{\alpha}_i = \alpha/k_o. \tag{19}$$

The third non-dimensional number is the elasticity per unit of mass, which can be formulated as

$$\hat{G} = G/\rho_i gh. \tag{20}$$

In the present research, $\hat{G}$ is called Elasticity number (which is inspired by Yu et al. (2019)). The other non-dimensional number is

$$\hat{\eta} = \eta/\rho_i \sqrt{gh^3}. \tag{21}$$

The fourth non-dimensional number is the relative density of the ice, and is calculated by

$$\hat{\rho} = \rho_i/\rho_w. \tag{22}$$

The hydrodynamic damping coefficient is normalized by

$$\hat{B} = B/\rho_w \sqrt{gh}. \tag{23}$$

The added mass coefficient can be normalized by

$$\hat{A} = A/\rho_w h^2. \tag{24}$$

Note that the added mass and damping coefficients are assumed to be two-dimensional.

## 3 Results and discussions

Results of wet beam models are presented in five separate sub-sections. The first sub-section presents a discussion on the effects of different parameters on the dispersion and decay rate plots. The primary aim is to provide a better understanding of the sensitivity of models to mechanical behavior of material and fluid forces. The second and third sub-sections discuss the ability of models in reconstruction of the decay rate and dispersion process through comparing their results against field and flume measurements. A brief summary of these measurements is documented in Table 1. In the fourth sub-section it is attempted to understand whether any other solid model can be used to formulate the dispersion relationship or not. Final sub-section of results presents a deep discussion on models and their abilities with covering limitations of models.

### 3.1 Effects of different parameters on decay rate and dispersion process

Figure 2 shows the normalized decay rates calculated by setting different values for the mechanical properties of the cover. Left and right columns respectively show the data found for KV and Maxwell materials. The dashed and dotted curves denote the results of dispersion relation of PE material (i.e., RP model with added mass contribution). The first and second rows show the effects of dynamic viscosity on the decay rate. The results presented in the first row correspond to a material with $\sqrt{\hat{G}} = 340$ (large flextral rigidity) and the ones plotted in the panels of second row correspond to a material with $\sqrt{\hat{G}} = 3.4$ (low flextral rigidity).

As seen, for a viscoelastic material with a larger flextral rigidity, $\hat{\alpha}$ increases under the increase in $\hat{\lambda}_o$ reaching a maximum value at a critical wavelength. With the increase in $\hat{\lambda}_o$, $\hat{\alpha}$ decreases. For a KV material, the critical $\hat{\lambda}_o$ is sensitive to the dynamic viscosity, growing with the increase in the dynamic viscosity. For a Maxwell material, the increase in the dynamic viscosity reduces the decay rate. Interestingly, the pure elastic model gives negligible decay rates at small dimensionless wavelengths ($\hat{\lambda}_o < 50$). The dimensionless fluid damping used to calculate the decay rates is set to be 0.0032.

To provide a clear picture of the effects of the fluid damping on the decay rates, $\hat{\alpha}$ vs. $\hat{\lambda}_o$ curve the pure elastic model gives is also plotted. The decay rates of a pure elastic material peaks at a specific $\hat{\lambda}_o$. Interestingly, the decay rates of the viscoelastic materials converge to that of the pure elastic model, signifying that the contribution of solid-based energy damping vanishes at relatively long waves. Instead, the fluid-based energy damping is dominant over the range of long waves. Note that the peak observed in $\hat{\alpha}$ vs. $\hat{\lambda}_o$ curves constructed using KV and Maxwell models is due to the method used for normalizing the decay

**Table 1.** Cases studied in the present paper.

| Reference | Type of cover | Type of test |
|---|---|---|
| Voermans et al. (2021) | Landfast ice covers (Arctic and Antarctic) | Field measurement |
| Wadhams et al. (1988) | Unconsolidated ice field (Greenland Sea) | Field measurement |
| Meylan et al. (2014) | Unconsolidated ice field | Field measurement |
| Yiew et al. (2019) | Freshwater ice cover | Flume measurement |
| Sree et al. (2018) | Viscoleastic cover | Flume measurement |

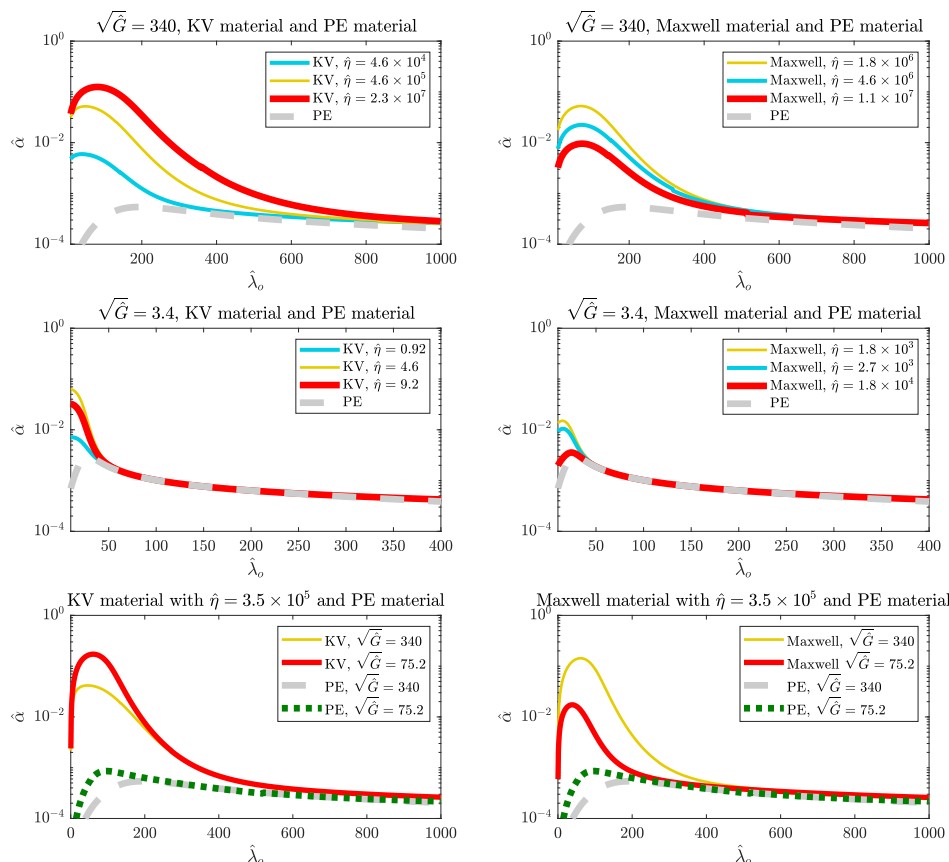

**Figure 2.** Effects of the dynamic viscosity (upper and middle rows) and Elasticity number (lower row) on $\hat{\alpha}$ vs. $\hat{\lambda}_o$ curves of different solid models give. Dashed and dotted models denote the decay rates that a PE material (RP model with consideration of added mass force) predicates.

rate, and will not emerge if dimensional data is plotted. The dimensional data is presented later and it will be shown that the decay rate of the pure elastic material peaks in short-wave range, though those of viscoelastic material does not.

Now we discuss the decay rates of covers with a lower rigid (second row). No critical $\hat{\lambda}_o$ is observed in $\hat{\alpha}$ vs. $\hat{\lambda}_o$ curves the KV model gives for a low Elasticity number. Decay rate of KV model with low rigidity reduces with an increase in dimensionless wavelength. The increase in the dynamic viscosity can affect the decay rate, though its effects are noticeable at shorter waves. For longer waves, however, different values of $\hat{\eta}$ give similar decay rates, which match with the those of a pure elastic material. This matches with what was observed for the larger rigidity (upper row). For a Maxwell material with low rigidity, a critical $\hat{\lambda}_o$ emerges over the short wavelength range. Similar to KV materials, the decay rate of Maxwell materials with low rigidity are sensitive to the dynamic viscosity over a very short range of wavelength. The $\hat{\alpha}$ vs. $\hat{\lambda}_o$ curves found by

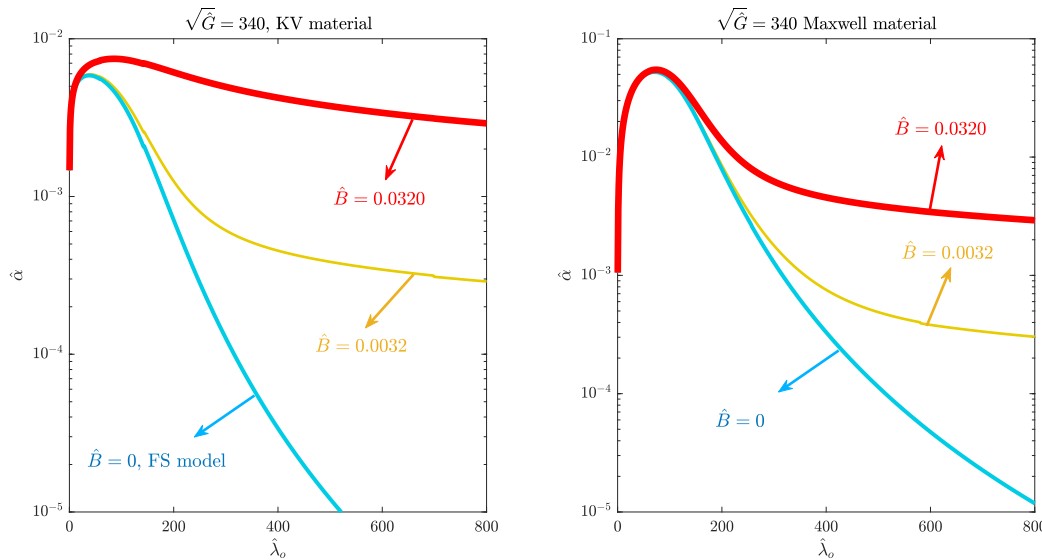

**Figure 3.** Effects of fluid damping coefficient on $\hat{\alpha}$ vs. $\hat{\lambda}_o$ curves viscoelastic models give.

setting different values for the dynamic viscosity converge to each other and finally aligned to the decay rates of pure model. This again confirms that fluid-based energy damping becomes dominant with the increase in the dimensionless wavelength.

The last row of Figure 2 compares the decay rates of materials with different Elasticity numbers. As apparent, when a KV model is used, an increase in Elasticity number reduces the decay rate. The most significant effects of elasticity on decay rate emerge at short wavelengths, where the solid-based energy damping is expected to be dominant. The decay rates of KV materials with different Elasticity numbers converge to those of the pure elastic material. The decay rate of a Maxwell material is proportional to its elasticity. Similar to decay rate plots of KV materials with different Elasticity numbers, decay rates of Maxwell materials with different Elasticity numbers converge to what the pure elastic model predicts. The $\hat{\alpha}$ vs. $\hat{\lambda}_o$ curves of pure elastic material with a larger flextural rigidity, $\hat{\lambda}_o$ peaks at a longer $\hat{\lambda}_o$. This means that, in case a RP model highly under-predicts the decay rate with setting a realistic elastic modulus, a much lower elastic modulus may be used for fitting the experimental data with the dispersion relationship. This has been observed by Mosig et al. (2015). They demonstrated that an elastic modulus of $\approx 3.2 \times 10^7 \ Pa$ gives the best fitting, though the shear modulus of ice is expected to be greater than $1 \ GPa$.

Figure 3 shows how the consideration of fluid damping can affect the decay rates. Three different plots are presented, each of which shows the decay rates found by setting different values for the fluid damping coefficient. The solid blue curves show the data found by setting a zero fluid damping coefficient, *i.e*, this plot represents the decay rates calculated in the absence of fluid damping. The two other plots show the decay rates predicted by setting two different values for the damping coefficient.

The left panel of Figure 3 shows the data related to a KV material. As apparent, when the fluid damping coefficient is set to be zero, the decay rate decreases with a very high rate at $\hat{\lambda}_o > 180$. By assuming a non-zero value for the damping coefficient, one abrupt reduction in $\hat{\alpha}$ vs. $\hat{\lambda}_o$ curve occurs, though the rate of the reduction of the decay rate as a function of the dimensionless

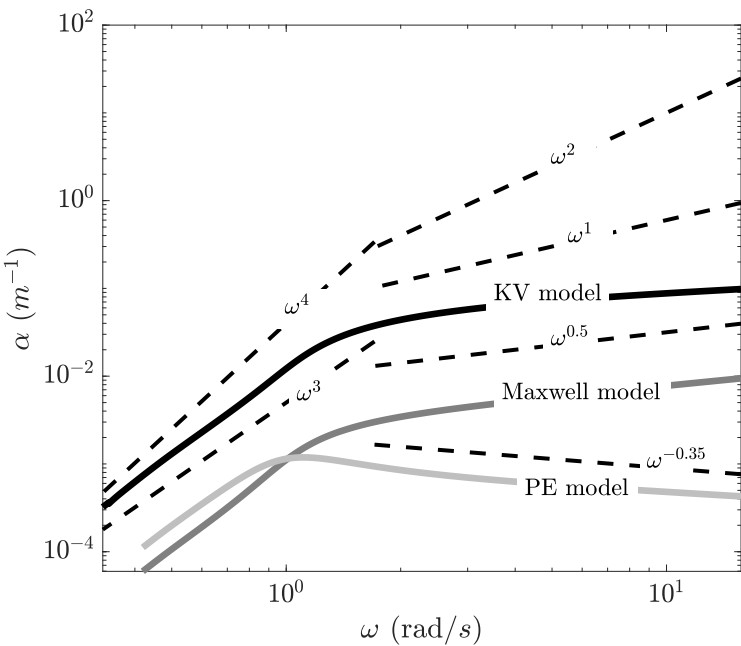

**Figure 4.** Log-log plots of $\alpha$ vs. $\omega$ curves constructed using different models. Curves are constructed by setting shear modulus to be $1\ GPa$.

wavelength decreases with the increase in the wavelength. Eventually the decay rate decreases with a mild rate over the range of $\hat{\lambda}_o > 400$. This confirms that the fluid-based energy damping becomes dominant over this range. For a large fluid-damping coefficient, as seen, the sudden reduction in the decay rate does not occur, and decreases with a low rate after reaching its peak value.

The decay rates of a Maxwell material with considering different damping coefficients are plotted in the right panel. Trends of the presented curves are consistent with the ones observed in the left panel of Figure 3 (KV material). This demonstrates that the effects of fluid-damping on decay rates are insensitive to the nature considered for materiel, which matches with the presented formulations for the dispersion relationships.

To understand the role of fluid-based damping and effects of viscosity on the decay rate more deeply, $\alpha$ vs. $\omega$ curves are plotted on a log-log scale. Dissimilar to previous Figures, the dimensional data is presented as it helps understand the dependency of $\alpha$ as a function of $\omega$. The $\alpha$ vs. $\omega$ curve corresponding to pure elastic material (RP model with added mass) peaks at a frequency of $\approx 1$ rad/$s$, though curves constructed using Maxwell and KV models do not peak. Note that the present data is dimensional, and if the dimensionless data (i.e. $\hat{\alpha}$) was presented, all curves would peak as it was observed before (see Figure 2).

$\alpha$ values predicted using the pure are proportional to $\omega^3$ in the long-wave regime. This matches with analysis presented by Meylan et al. (2018). In the short-wave range, decay rate of pure elastic material is proportional to $\omega^{-0.35}$. We recall that it was

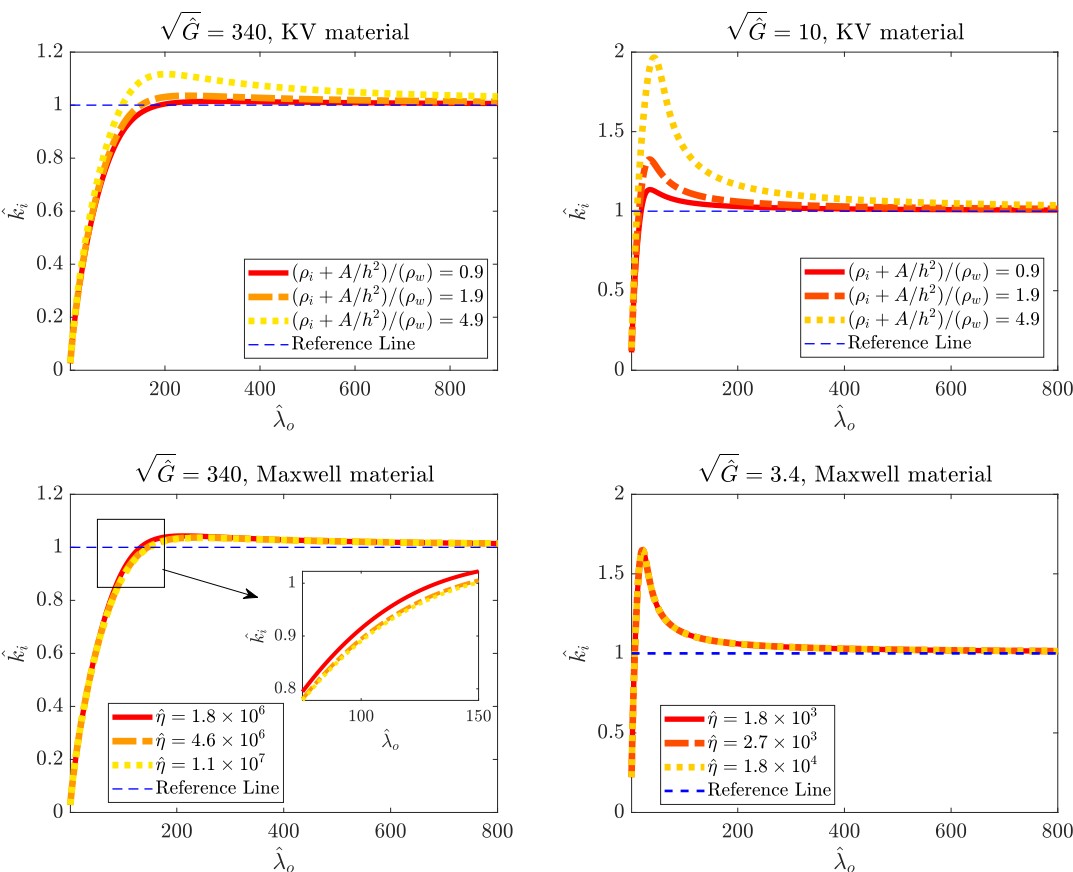

**Figure 5.** Effects of the added mass force (upper row) and the dynamic viscosity (lower row) on the dispersion curves of waves propagating in an ice-cover.

discussed that one solution to match predictions of RP model with experimental data over short range wave is to decrease the shear modulus of material, which may significantly widen the zone over which $\alpha$ is proportional to $\omega^3$.

The $\alpha$ vs. $\omega$ curves of viscoelastic materials are approximately proportional to $\omega^3$ over a wider range of frequency as compared to the curves constructed using the pure elastic model. In the short-wave zone, $\alpha$ values predicted using the KV model are proportional to $\omega^n$, where $n$ is slightly lower than $0.5$. Overall, it can be concluded that consideration of solid-based damping widens the frequnecy range over which $\alpha \propto \omega^3$, and cancels out decrease of decay over short-wave range.

The dispersion relationships are also employed to calculate wavenumbers of waves propagating into viscoelastic covers. Results are depicted in Figure 5. The density of covers is set to be $\rho_i/\rho_w = 0.9$ which is very close to that of the sea ice. The two upper panels of Figure 5 show the dimensionless wavenumbers found for KV materials with different Elasticity numbers. Zero added mass coefficients (solid curve) and two non-zero added mass coefficients are considered.

As apparent, when the Elasticity number of a KV material is greater (left) and the added mass is nil, the dimensionless wavenumber is below 1.0 at small values of $\hat{\lambda}_o$. This means that, compared to an open-sea condition, gravity waves traveling into a solid cover becomes longer over the range of short open-water wavelength. $\hat{k}_i$ increases with the increase in $\hat{\lambda}_o$. and eventually converges into 1.0, where plotted curves are flattened out, meeting the reference line (the dashed blue line). Interestingly, added mass can affect the wavenumber. Under the action of added mass force, wavenumber becomes greater. The effects of added mass on the wavenumber are more significant over $100 < \hat{\lambda}_o < 600$. When the added mass coefficient is set to be very large, the dimensionless wavenumber becomes greater than 1.0, reaching a maximum value and then decreases, converging into 1.0. The same conclusion can be made for a pure elastic model as dispersion process of pure elastic material and KV model are similar.

For a KV material with low density and a zero added mass coefficient, the wavenumber is below 1.0 when the dimensionless open-water wavelength is small. Compared to a KV material with a greater Elasticity number (left panel), $\hat{k}_i$ becomes lower than 1.0 over a narrower range of $\hat{\lambda}_o$. This implies that waves propagating into a solid cover are lengthened over a wider range of dimensionless open-water wavelength when rigidity increases. $\hat{k}_i$ becomes greater than 1.0 and reaches a peak value over the range of short the open-water wavelength, which is in contrast with what was observed for material with a larger Elasticity number and zero added mass (left panel).

The added mass force can affect the dispersion process of waves propagating into cover with lower Elasticity number by increasing the wavenumber, though its influences on the waves advancing in the cover with lower Elasticity number are more noticeable compared to cover with larger Elasticty Number.

The two lower panels of Figure 4 show the calculated dimensionless wavenumbers of gravity waves propagating into a solid cover hypothesized to behave in the same way Maxwell materials do. Left and right panels respectively show the curves corresponding to covers with relatively large and low Elasticity numbers. The trends of curves plotted in the left panel confirm that the dynamic viscosity can slightly affect the wavenumber of waves advancing in the cover with a large Elasticity number. The influence of dynamic viscosity on wavenumber is not significant and emerges over a relatively narrow range, (the close-up view provides the evidence). Dimensionless wavenumber is seen to be insensitive to the dynamic viscosity of Maxwell materials when Elasticity number is low (right panel).

## 3.2 Ability of models in prediction of the decay rates

This section presents comparisons between predictions of the models against decay rates found through field and flume measurments. In all runs, the $A/\rho_w h^2$ is set to be 1.0. Decay rates of two recent field tests are presented in Figure 6 (circle markers). Upper and lower rows respectively display the data corresponding to field tests took place in Arctic and Antarctic. The first, second, and third columns of Figure 6 respectively show the decay rates predicted by prescribing Kelvin-Voigt, Maxwell, and pure elastic materials. The decay rates found by setting zero damping coefficient are also plotted to demonstrate how the inclusion of the fluid damping can improve the accuracy of models in the prediction of the decay rates.

Viscoelastic models cannot follow the field data when fluid damping is set to be zero, and tails of curves diverge from that of the field data. This can be seen in both upper and lower rows of Figure 6. In this condition, the increase of the dynamic viscosity

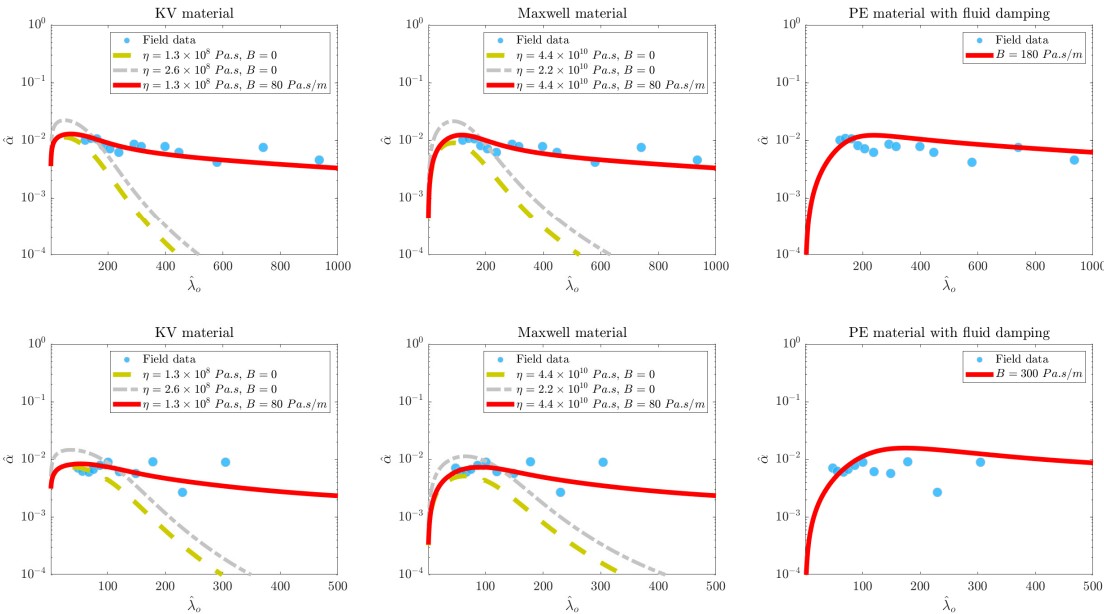

**Figure 6.** Comparisons between $\hat{\alpha}$ vs. $\hat{\lambda}_o$ curves predicted by different models and the data measured by Voermans et al. (2021). Upper and lower rows respectively show data measured in Arctic and Antarctic. In the first and second columns, decay rates predicted by setting a zero fluid damping are also plotted. The shear Modulus of ice is set to be $1\ GPa$.

cannot affect the trend of $\hat{\alpha}$ vs. $\hat{\lambda}_o$, and may only shift the curve upward or downward, depending on the nature considered for the material. But, when the fluid damping coefficient is set to be non-zero, the $\hat{\alpha}$ vs. $\hat{\lambda}_o$ follow the field data. This confirms that fluid damping can contribute in energy damping occurring under a landfast ice cover. Interestingly, fluid damping coefficients used to predict the decay rates of viscoelastic are similar.

For the KV material, a dynamic viscosity of $\approx 1.3 \times 10^8 Pa.s$ gives the best fitting for both ice covers. But for a Maxwell 330 material, a dynamic viscosity of $\approx 4.4 \times 10^{10} Pa.s$ gives the best fitting. Finally, the pure elastic material assumption can also be used to match curves with the field experiments. But, compared to KV and Maxwell models, a greater value of fluid damping coefficient gives the best fitting for an elastic model.

The decay rates of two different broken ice fields are calculated using the wet beam models and are compared against measurements. Field data and calculated decay rates are plotted in Figure 7. The data presented in upper and lower panels of 335 this Figure are respectively documented in Wadhams et al. (1988) and Meylan et al. (2014).

Wet beam models established for viscoelastic materials can predict decay rate curves of broken ice field with a good level of accuracy when fluid damping is incorporated in calculations. Similar to curves plotted in Figure 6, increase or decrease in the dynamic viscosity of the material can shift the curves vertically, not affecting the tail of $\hat{\alpha}$ vs. $\hat{\lambda}_o$ curve. The curve given by a pure elastic material is not accurate at all dimensionless wavelengths. This implies that to accurately compute the decay

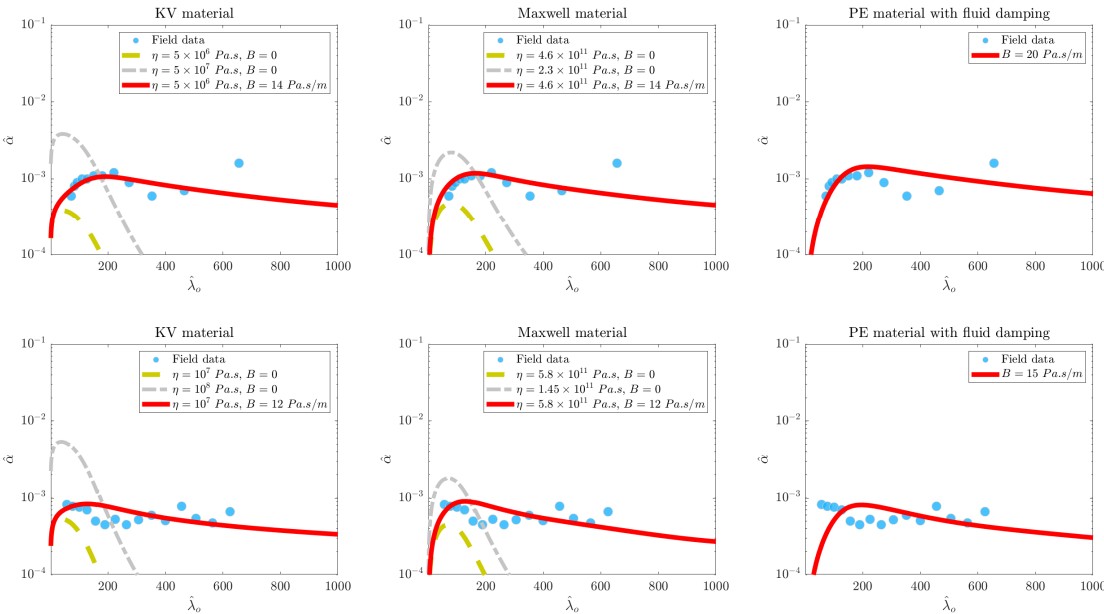

**Figure 7.** Comparisons between $\hat{\alpha}$ vs. $\hat{\lambda}_o$ curves predicted by different models and the data measured by Wadhams et al. (1988), upper row, and Meylan et al. (2014), lower row. In the first and second columns, decay rates predicted by setting a zero fluid damping are also plotted. The shear Modulus of ice is set to be $1\ GPa$.

rate of a broken ice field, solid-based energy damping caused by the viscoleastic behaviour of the solid needs to be taken into consideration which is being lacked in a pure elastic model. The dimensional data and curves related to field tests documented in Wadhams et al. (1988) are presented in Appendix A. The aim is to show whether a peak may emerge in $\alpha$ vs. $T$ curves when models are used or not, as emergence of a peak value in high-frequency range may not be real (Thomson et al. (2021)).

     Wet beam models are employed to calculate the decay rates of waves traveling into viscoelastic covers, with an aim to

understand whether they can be used to predict attenuation rates measured in flume tests or not. The reconstructed $\hat{\alpha}$ vs. $\hat{\lambda}_o$ curves are presented in Figure 8. The upper and lower rows show the data corresponding to covers with low and large rigidity.

     The curves reconstructed by viscoelastic models cannot follow the experimental data if the fluid damping is not considered. Dashed and dashed-dotted curves plotted in the first and second columns provide shreds of evidence. Their results only match with experimental data at very short dimensionless open-water wavelengths. Viscoelastic-based models can accurately predict

decay rates when the fluid damping is set to be non-zero. The elastic model can calculate decay rates with an acceptable level of accuracy. The fluid damping that gives the best fitting for pure elastic material is slightly greater than those of models built for viscoelastic materials.

     At the final stage, the decay rates of the freshwater ice formed in the wave flume of the University of Melbourne are predicted by using the presented models. Figure 9 displays the decay rates plots and experimental data. The results presented in the upper

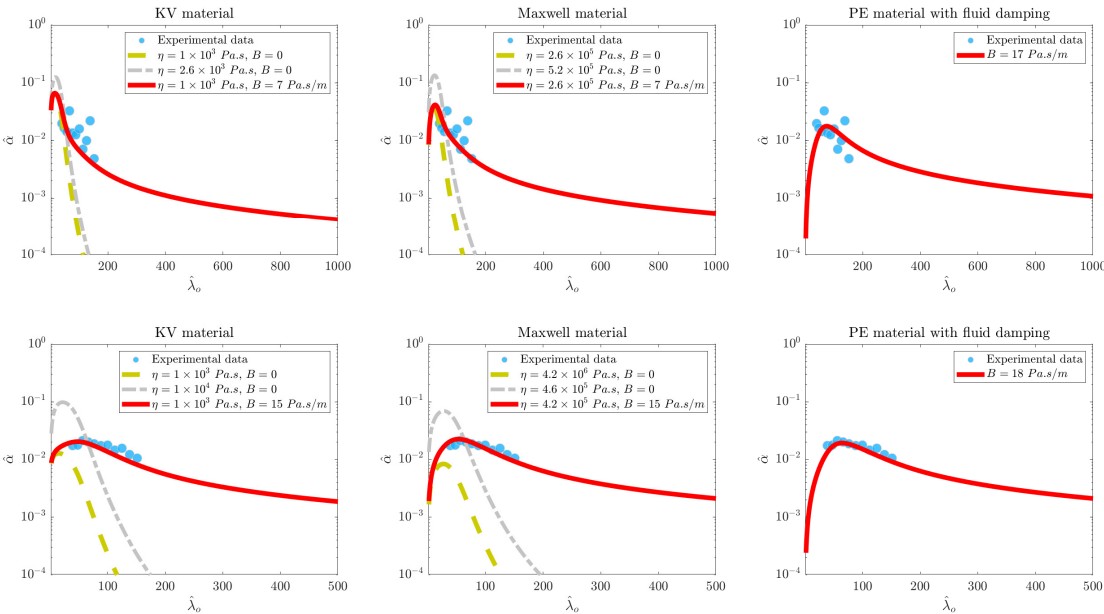

**Figure 8.** Comparison between $\hat{\alpha}$ vs. $\hat{\lambda}_o$ curves predicted by models and the data measured by Sree et al. (2018). Upper and lower rows show the data related to a cover with a shear Moduli of $20\ KPa$ and $80\ KPa$ respectively.

and lower rows of Figure 9 are related to $1\ cm$ and $1.5\ cm$ thick ice covers. Before conducting flume experiments, Parra et al. (2020) measured the Young Modulus of the dry freshwater ice, reporting that Young Modulus is $\approx 3\ GPa$.

The KV model can accurately predict the decay rates of $1\ cm$ thick ice when dynamic viscosity is set to be $2.6 \times 10^9\ Pa.s$. Inclusion of the fluid damping is seen to be ineffective when KV model is used. The Maxwell model can also predict the decay rate curve of the $1\ cm$ thick ice cover with a good level of accuracy. But the decay rates Maxwell model predicts depend on the fluid damping. The model developed for pure elastic materials can construct $\hat{\alpha}$ vs. $\hat{\lambda}_o$ curve if fluid damping is set to be $1300$ $Pa.s/m$, which is as $13$ times greater than those considered for KV and Maxwell models. This significant difference between fluid damping of pure elastic model and viscoelastic models, which was never observed in previous Figures, indicating that the solid-based damping highly contributes to energy dissipation over the range of tested waves in the laboratory. To compensate the absence of the solid-based energy damping of the pure elastic model, a very large fluid damping coefficient needs to be used. This is in contrast with what was observed for the tests of Sree et al. (2018), where the difference between fluid damping of elastic and viscoelastic models was not significant. The big difference between the Elasticity numbers of the covers tested by Sree et al (2018) and ice covers tested by Yiew et al. (2019) explains this behavior. It was also demonstrated that when Elasticity number is low, the solid-based energy damping contributes to energy dissipation over a narrow range of wavelengths (second row of Figure 2), compared to a cover with a larger Elasticity number (first row of Figure 2).

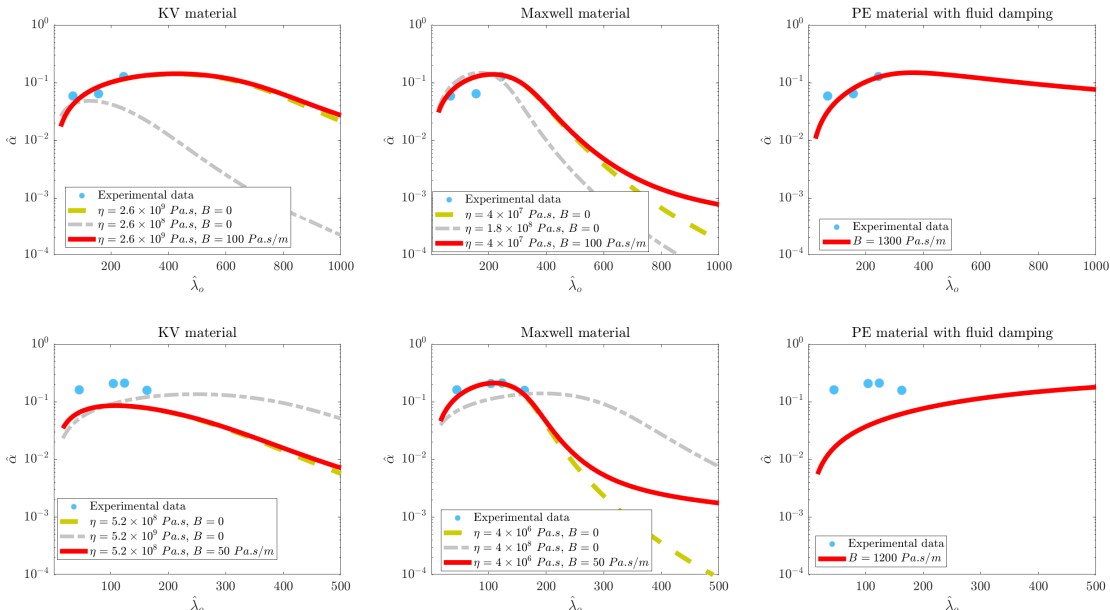

**Figure 9.** Comparison between $\hat{\alpha}$ vs. $\hat{\lambda}_o$ curves predicted by models and the data measured by Yiew et al. (2019). Upper and lower rows show the data related to freshwater ice covers with thicknesses of $1\ cm$ and $1.5\ cm$, respectively.

The KV model under-predicts the decay rate curve of $1.5\ cm$ thick ice cover regardless of the values of the dynamic viscosity and the fluid damping coefficients. The Maxwell model, however, can construct $\hat{\alpha}$ vs. $\hat{\lambda}_o$ curve when a dynamic viscosity value of $4 \times 10^6\ Pa.s$ is set. This value is lower than that the dynamic viscosity gave the best fitting for the thinner ice. Artificial effects, boundary conditions, and the presence of side walls may cause a larger energy damping pattern when fluid interacts with the thicker ice (Sutherland et al., 2016). Thus, different values of dynamic viscosity give the best fitting for $1\ cm$ and

$1.5\ cm$ thick ice covers. It is important to note that dynamic viscosity values giving the best fitting for the landfast ice were not seen to be different (Figure 6), where artificial effects are less likely to contribute. As seen, fluid damping does not have any significant effect on the curve over the range of tested waves. It can only affect the tail of curve. This again confirms that the solid-based energy damping is dominant over the range of tested waves. If tests covered a wider range of open-water wavelengths, especially longer ones, a proper value for fluid damping could be found through fitting predicted curves with

experimental data. When the pure elastic model is used to calculate decay rates, a very large fluid damping coefficient needs to be set, but the curve can never meet the experimental data.

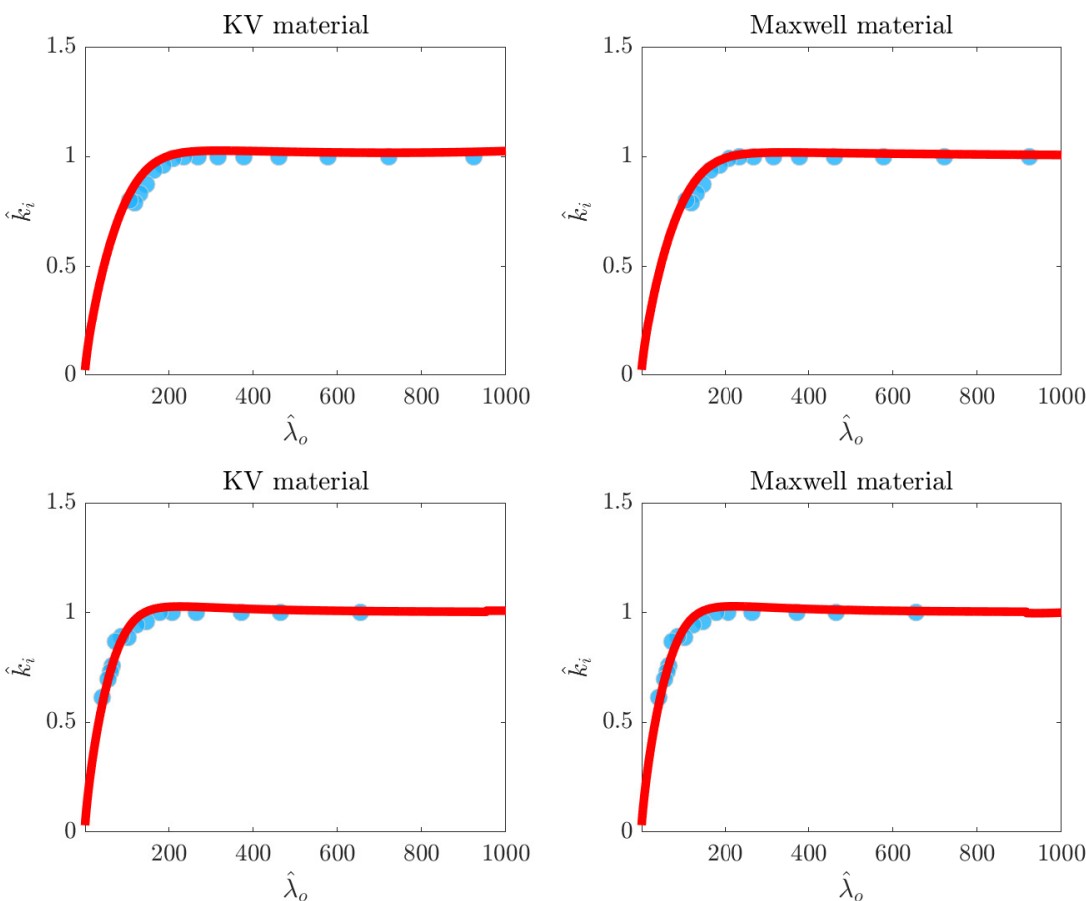

**Figure 10.** Comparison between $\hat{k}_i$ vs. $\hat{\lambda}_o$ curves predicted by wet beam models and the data measured by Voermans et al. (2021). Upper and lower rows respectively show data related to tests performed in Arctic and Antarctic. The shear Modulus of ice is set to be $1\ GPa$.

### 3.3 Ability of models in the prediction of the dispersion process

The accuracy of models in the calculation of the dispersion process of waves travelling through viscoelastic covers is also evaluated. First, the dispersion curves of waves propagating into landfast ice are constructed, and then the dispersion curves of
385 waves advancing in freshwater ice are plotted.

Figure 10 shows the normalized wavenumber $\hat{k}_i$ vs. $\hat{\lambda}_o$ curves of landfast ice. Symbols indicate the field measurements and curves denote the calculated dimensionless wavenumbers. Left and right panels respectively show predictions made for KV and Maxwell materials. As seen, both models can predict the dispersion process of waves traveling into the landfast ice with an acceptable level of accuracy. The inputs that are used for the construction of dispersion curves are similar to what gives the
390 best fitting for decay rates.

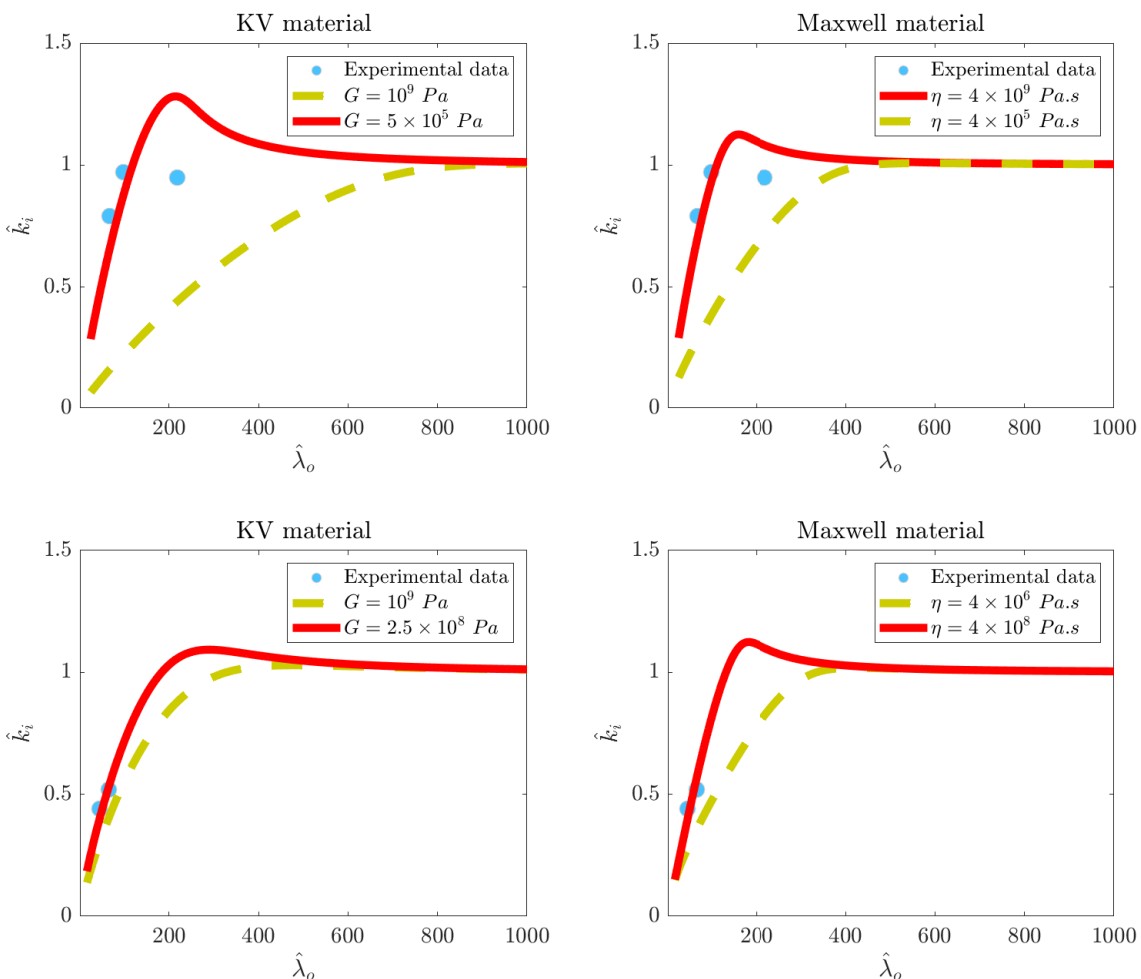

**Figure 11.** Comparison between $\hat{k}_i$ vs. $\hat{\lambda}_o$ curves predicted by wet beam models and the data measured by Yiew et al. (2019). Upper and lower rows show the data related to freshwater ice covers with thicknesses of $1\ cm$ and $1.5\ cm$, respectively.

Figure 11 shows the $\hat{k}_i$ vs. $\hat{\lambda}_o$ plots of the flume tests performed in University of Melbourne. It was observed that none of the models can predict the dispersion process of waves interaction with freshwater ice with the inputs gave the best fitting for the decay rates plots (Figure 9). This is in contrast with what was observed in Figure 10, where models accurately predicted the wave dispersion process with similar inputs utilized for the calculation of decay rates. As seen, when dispersion process of waves interacting with freshwater is constructed by using a KV model, Young Modulus values of $5 \times 10^5\ Pa$ and $2.5 \times 10^8\ Pa$ give the best fitting for the dispersion processes of $1\ cm$ and $1.5\ cm$ ice covers, respectively . As explained in the Introduction Section, some other researchers observed that the effective Young Modulus of a material is different and smaller from what

is measured in dry tests. For a Maxwell material, however, different values of dynamic viscosity can be used to fit the curves with the experimental data. It was previously demonstrated that the dynamic viscosity can affect $\hat{k}_i$ vs. $\hat{\lambda}_o$ plots by increasing the wavenumber over the range of short waves. By setting larger values for the dynamic viscosity, the constructed dispersion curve matches with the experimental data (the solid curve).

Note that wavenumbers of field tests documented in Wadhams et al. (1988) and Meylan et al. (2014) are not presented. Therefore, we were not able to compare capability of wet beam models in the predictions of wavenumbers of the field with broken ice against those of a broken ice field.

### 3.4 Other models

In sub-sections 3.2 and 3.3, it was observed that proposed models cannot accurately calculate the dispersion process and decay rates of waves advancing into the freshwater ice. Artificial effects may have contributed to the dispersion and dissipation process, though the large difference between inputs giving the best fit for the decay rate and dispersion plots still leaves us with a big question mark about the main reason for such a difference.

As discussed, other researchers have hypnotized that the effective elasticity or dynamic viscosity of the wet ice interacting with water waves can be different from what have been measured in dry tests. While the whole paper looks into ability of KV, Maxwell in the prediction of decay rates and dispersion process and discusses how accurate their results can be, the idea of developing other models by considering various viscoelastic materials rises. We can use the other solid models to evaluate whether Young Modulus giving the best result for decay rates and dispersion process match with what was found in dry tests or not.

To provide an answer to the above question, the linear combination of KV and Maxwell materials, known as Standard Linear Solid (SLS) model, is used to describe the mechanical behaviour of ice. Two spring elements ($G_1$ and $G_2$) are used to formulate these models. Standard Linear Solid models are also called "Zener". Two different Zener models have been introduced in this paper. The first one is a SLS model established using a Kelvin approach (a spring element in series with a KV arm, also known as the first order generalized Kelvin-Voigt model). The other model is acquired employing a Maxwell approach (a spring element in parallel with a Maxwell arm, also known as the first order generalized Maxwell model). The schematic of both of

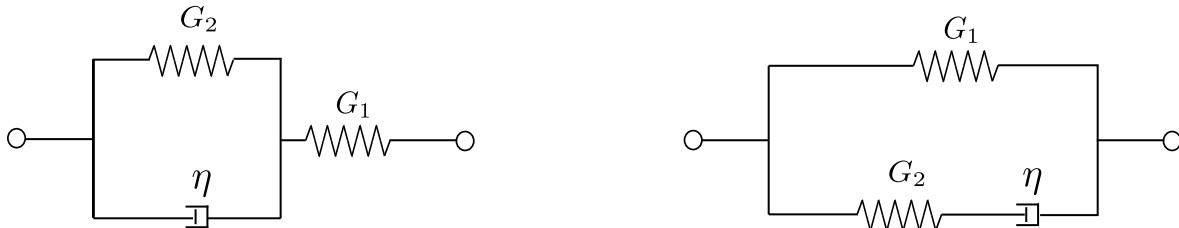

**Figure 12.** Standard linear solid models. Left and right panels respectively show the SLS KV and SLS Maxwell models.

these models are shown in Figure 12. The storage and loss moduli of these two solid models are formulated as

$$G_E = G_0\big(1 - \frac{p}{1+\tau^2\omega^2}\big) - iG_0\frac{p\tau\omega}{1+\tau^2\omega^2}. \tag{25}$$

For the SLS KV material, $G_0$, $p$ and $\tau$ are given by

$$G_0 = G_1, \qquad\qquad p = \frac{G_1}{G_1+G_2}, \qquad\qquad \tau = \frac{\eta}{G_1+G_2}. \tag{26}$$

For the SLS Maxwell material, $G_0$, $p$ and $\tau$ are calculated as

$$G_0 = G_1+G_2, \qquad\qquad p = \frac{G_2}{G_1+G_2}, \qquad\qquad \tau = \frac{\eta}{G_2}. \tag{27}$$

Readers who are interested in the above formulations are referred to Serra-Aguila et al. (2019). The same way used to establish dispersion relationships in the Section 2 is applied, and the related dispersion relationship for SLS materials is formulated as

$$\omega^2 = \left(\frac{G_0\big(1-\frac{p}{1+\tau^2\omega^2}\big)h^3}{6(1-\nu)\rho_w}k^4 - i\frac{G_0\frac{p\tau\omega}{1+\tau^2\omega^2}h^3}{6(1-\nu)\rho_w}k^4 + g - i\omega\frac{B}{\rho_w} - \frac{\rho_i(h+A)}{\rho_w}\omega^2\right)k\tanh(kD), \tag{28}$$

Decay rates and dispersion of waves propagating into the $1.5\ cm$ freshwater ice cover are recalculated by using the two introduced SLS models and plotted in Figure 13. The Left and right panels respectively show the results found using SLS KV and SLS Maxwell models. Dispersion curves are plotted in upper panels, and the decay rates are illustrated in lower panels. SLS KV and SLS Maxwell models both construct $\hat{k}_i$ vs. $\hat{\lambda}_o$ plots similarly. Interestingly, the equivalent Young Modulus of the SLS KV model is $2.5\times 10^4\ Pa$ and that of the SLS Maxwell model is $3.3\times 10^9\ Pa$. This confirms that the SLS KV model which is built using the KV approach can only predict the predicted process with a very low equivalent Young Modulus, which is much smaller than Young Modulus found in dry tests (which is around $2.6\times 10^9\ Pa$). But the equivalent Young Modulus of the SLS Maxwell model is close to what is found in dry tests. This signifies that the freshwater ice formed in the flume is more likely to behave similarly to a SLS Maxwell material. This can explain the difference between the effective Young Modulus reported in different experimental researches. Researchers who conducted those flume/basin experiments concluded that a lower Young Modulus should be used to calculate the dispersion process of waves propagating into ice when a dispersion equation built on the basis of pure elastic material (or the KV material) is employed. But as observed here, the material is more prone to show a rheological behavior falling in between those of the KV and Maxwell materials, arranged using a Maxwell approach, the equivalent Young Modulus of which is close to what is measured in dry tests.

The decay rates are seen to be well predicted by both models. The interesting point is that, the KV model was found not be able to construct the decay rates of $1.5\ cm$ thick. But as seen here, the results of SLS KV model fairly agree with experimental measurements. The SLS Maxwell model can predict the decay rate very well. The trend of the decay rates given by SLS Maxwell and SLS KV are very similar.

Note that adding more elements may make the dispersion relationship more complex, leaving us with more options (i.e., more inputs), though it can also lead to over-fitting as more parameters are needed to be tuned. In the future, more studies can

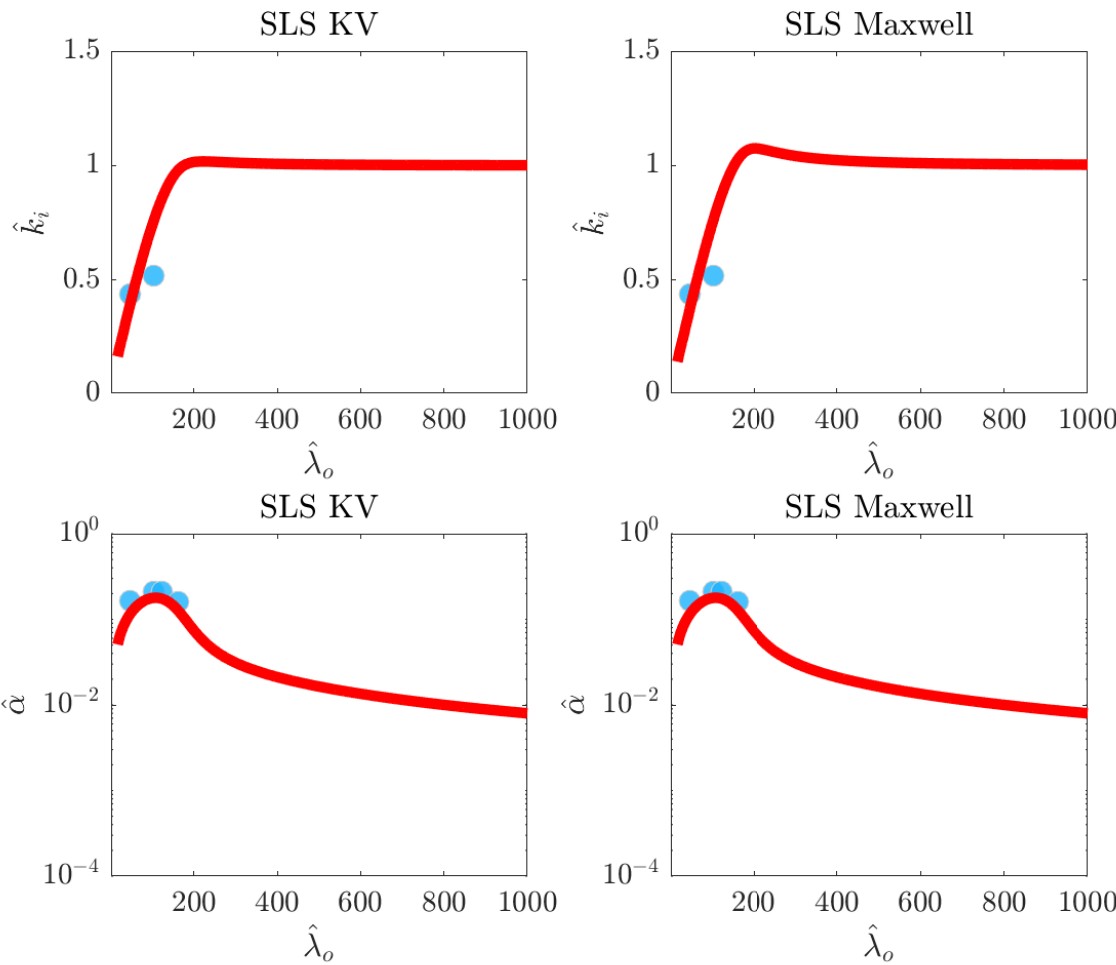

**Figure 13.** Comparison between predictions of SLS models and values measured by Yiew et al. (2019). Upper and lower rows show the decay rates and dispersion plots. Equivalent Young Modulus of the SLS KV model is $2.5 \times 10^4$ $Pa$ and equivalent Young Modulus of the SLS Maxwell model is $3.3 \times 10^9$ $Pa$. Dynamic viscosity of SLS KV model is $4 \times 10^7$ $Pa.s$ and dynamic viscosity of the SLS Maxwell model is $3.5 \times 10^9$ $Pa.s$.

be carried out to understand the mechanical behavior of different types of viscoelastic ice, and it can be investigated whether it is required to use solid models with more than two elements or not.

### 3.5 Final discussion

Decay rates and dispersion curves of gravity waves propagating into an ice cover can be constructed using the presented wet beam models. With the same setups (e.g. dynamic viscosity and Young Modulus), different models can give different curves

for the decay rates and the dispersion process. Thus, the choice of the model for the prediction of the decay rates and the dispersion process is very important. This needs a better understanding of the ice mechanics and the way it is formed. Simply stated, the mechanics of ice is a very complicated field of research, with lots of open questions not have been answered yet.

Models formulated in the present research follow the basis of the Euler-Bernoulli beam theory, which describes displacements of a solid layer by assuming small motions. To provide a clear picture of this theory, transverse sections of a beam flexed due to an external/internal load should be assumed. If the displacement field follows the Euler-Bernoulli law, the normal vector of any transverse section is always parallel to the axis of the beam. That is, no local rotational motion occurs. This is different from what happens for a solid body following the Timoshenko–Ehrenfest beam theory, where rotational motions are taken into consideration (Timoshenko and Woinowsky-Krieger, 1959). Furthermore, to establish any model, the mechanical behaviour of a substance should be formulated. It left us with two common choices as we are following a beam theory: The ice can be hypothesized to be either elastic or viscoelastic (we may have other choices for a solid ice layer, example: poroelastic ice). When elastic behavior is considered, Hooke's law is utilized to formulate the relationship between the stress and the resulting strain. If viscoelastic behavior is assumed, different linear models can be used. The KV and Maxwell models are commonly used for this aim, the former which represents a viscoelastic solid body, and the latter which represents a mass of viscoelastic fluid.

Using mechanical behavior prescribed for the ice and employing an Euler-Bernoulli equation, dispersion relationships can be formulated. If the ice layer is assumed to be dry and no fluid-solid interaction is taken into account, dispersion relationships for wave motions in dry beams are acquired. If the multi-physical problem is considered, the ice layer is assumed to settle down on the water surface and fluid forces emerge. If the fluid motion is assumed to be linear, a fluid damping force can also be included in the beam motion equation, as was suggested by Robinson and Palmer (1990). The original model of Greenhill (1886) and Fox and Squire (1991) do not consider fluid damping. Fluid damping was considered in some other models, which were formulated for an elastic ice cover as dissipation was either formulated through introducing a complex term in flexural rigidity of the ice cover (solid-based energy damping) or by considering a linear damping term (fluid-based energy damping, Squire et al. (2021)). But, the fluid damping and the solid damping can be considered at the same time, leading to a more general dispersion relationship, compared to available ones.

In the present research, it is hypothesized that energy dissipation is caused by solid-based energy damping and fluid-based energy damping. The former is dominant over high frequencies (corresponding to short waves in an open-water condition), and the latter is dominant over small frequencies (corresponding to long waves in an open-water condition). The solid-based energy damping is caused by the viscoelastic resistance emerging in the solid and is absent for an elastic material. Regardless of the viscoelastic model used to treat the mechanical behavior of the material, the solid-based energy damping decreases with an increase in the wavelength. With an increase in Elasticity number, the wavelength range over which solid-based energy damping is dominant becomes wider. This matches with physics. The body is more rigid and solid responses can contribute to energy damping of longer waves, compared to bodies with low rigidity. The fluid-based energy dissipation is generated by a velocity dependent force, decreasing with the increase in wavelength. The energy damping triggered by a pure elastic cover is only due to the presence of fluid damping force. Thus, to compensate for the lack of solid-based energy damping of an

elastic cover, larger fluid damping coefficient needs to be considered. Energy decay triggered by the landfast ice covers which have large rigidity can be computed using the viscoelastic models if fluid damping is included. If it is not, models cannot work properly and the predicted decay curves diverge from field measurements. This well confirms that fluid-based energy damping contributes to total energy dissipation. Both viscoelastic models were seen to construct the decay curves, fairly follow field measurements. But the dynamic viscosity values were seen to be much different. A KV material may be a more realistic indicator of ice behavior as the lanfast ice is expected to be solid.

Decay rates of broken ice fields were seen to be calculated by both viscoelastic models when the fluid-based energy damping is considered. This provided us with another piece of evidence for the contribution of fluid damping coefficient force to energy dissipation. Compared to landfast ice covers, different values of the dynamic viscosity were observed to give the best fitting for the decay rates of waves advancing in water partially covered with broken ice floes. The coexistence of open-water and ice floes on the upper layer of the fluid domain can explain this. Models are formulated for an integrated layer of solid ice. If water is included, a mixed thin layer represents the cover. A volume fraction model may describe the dynamic viscosity of the layer. The dynamic viscosity of the water is much smaller compared to that of ice, but the water entrapped between ice floes may be turbulent, leading to eddy generation (turbulent flow can cause damping of waves, *e.g.* Tavakoli et al. (2022)). In some mathematical models, an artificial damping term is usually introduced to incorporate the gap effects on fluid pressure (Lu et al. (2010)). The KV model gives the best fitting of decay rate when the dynamic viscosity is reduced, compared to the landfast ice. In contrast, the Maxwell model gives the best fitting with a larger dynamic viscosity. Thus, what can be concluded is that if water reduces the dynamic viscosity of the upper layer, the KV model will more likely to be a better indicator of the ice behavior. Otherwise, the Maxwell model prescribes the mechanical behaviour of the material. In addition, in a broken ice field, the gap between ice floes may lead to emergence of an extra damping mechanism, which is introduced as an artificial damping in studies highlighting interactions between floating bodies. In the present research, the aim was to develop a model for an integrated viscoelastic ice cover with incorporation of fluid damping, and thus the artificial damping was not employed.

The decay rates of the freshwater ice were also calculated. The pure elastic materials can reconstruct the decay rates only if a very large fluid damping coefficient is used, which may not be realistic. But, viscoelastic models were seen to predict the decay rate plots. Models, however, were not able to capture the dispersion process under freshwater ice covers with the same input observed to give the fitted decay rates. Effective values were seen to construct dispersion plots with an acceptable level of accuracy. This has been observed and reported by other scholars who measured the wavelength and phase speed of disintegrated elastic/viscoelastic covers. The interesting point is that, when a Maxwell model is used, the dynamic viscosity can affect the dispersion process. This motivates us to build other models which formulate the storage modulus by applying the dynamic viscosity. Two available linear models were introduced. One is SLS KV and the other is SLS Maxwell. Both models include two springs and one dashpot. The equivalent Young Modulus giving the best fitting when SLS KV is used is $\approx 2.5 \times 10^4 \ Pa$, which is much smaller compared to that of real freshwater ice. But, for SLS Maxwell, the equivalent Young modulus was seen to be $\approx 3.3 \times 10^9 \ Pa$ which is close to that of freshwater ice. While it is very interesting to add more elements to the solid model, over-fitting may happen as more elements are needed to be tuned. This leaves us with another question: Is it required to consider more elements in the solid model or the effective shear modulus of the ice formed in the flume has been

highly reduced because of the ice formation process (for example, porosity may reduce effective shear modulus, Zong (2022))?

Based on what was observed when SLS models were used, it can be concluded that two-parameter models such as Maxwell and KV with realistic Young modulus, cannot reproduce mechanical behavior of the freshwater ice formed in the flume, and its behavior is different from that of the landfast ice. There are still doubts about the behaviour of broken sea ice. Both Maxwell and KV models were found to give the best fitting. One reason is that the wave phase is not measured in most of the field tests that took place in the broken ice field, and researchers were mostly concerned with decay rates. Since the phase speed and dispersion plots are not available, the performance of models in the reconstruction of dispersion plots cannot be evaluated. Especially, this could show whether changes in the dynamic viscosity can modify the accuracy of the Maxwell model in the prediction of dispersion or not (dispersion curves Maxwell model give is sensitive to dynamic viscosity). That may be still an open question for the future.

## 4   Conclusions

The common approach used to predict decay rates and dispersion of waves penetrating an ice cover is to formulate a dispersion relationship of a continuum medium, the roots of which give the decay rate and the relative wavenumber in an ice-covered sea. The majority of models developed for wave-ice interaction have been developed based on two common approaches. First, ice was assumed to be a viscous fluid or a viscoelastic solid layer. Second, ice was assumed to be elastic, and a fluid damping term is used to calculte the decay rate. This paper aimed to present wet beam models by considering both of fluid-based and solid-based energy dissipation mechanisms by accommodating different rheological for the ice layer (KV and Maxwell). Thus the models were called *"wet beam"* which refers to water-based forces that are taken into consideration.

Predictions of viscoelastic models and field measurements were quantitatively compared against each other. KV and Maxwell models were seen to reconstruct the decay rates and dispersion process of landfast ice with a great level of accuracy. Decay rates were observed to be poorly predicted if the fluid-based energy damping is not taken into account, suggesting that this mechanism has a very important role in ice-induced energy decay over the range of long waves. The decay rates of unconsolidated ice fields were also seen to be accurately predicted by KV and Maxwell models by setting a non-zero value for the fluid damping coefficient.

Decay rates and wave dispersion of freshwater ice were predicted. The pure elastic model was seen to predict the decay rate with an unrealistic fluid damping coefficient. The decay rates predicted by KV and Maxwell models were seen to agree with experiments, though the dispersion plots were observed to diverge from the experimental data.

Two standard linear solid models were used and two other dispersion relationships were formulated. These relationships were seen to predict the attenuation rate and the dispersion plots with a good level of accuracy. But, the SLS model which was fundamentally based on the KV material gave the best fitting with an unrealistic Young Modulus.

Overall, the wet beam dispersion relationships developed by accommodating two-parameter solid models are able to predict the decay rates and dispersion process of ice fields. But, for freshwater ice flume, a three-parameter solid model may increase

the accuracy of the predictions. As wet beam models were observed to be capable of fitting decay rates of different field measurements, they can be employed in wave spectral modeling of polar seas in the future, and also can be coupled with ice break-up models to simulate the evolution of Marginal Ice Zones. In the future, nonlinear models can also be developed to consider the effects of wave steepness on dispersion, and other beam theories can be employed. In addition, viscosity and turbulence effects, can also be Incorporated into wet beam models.

*Video supplement.* This research does not include any supplementary file.

## Appendix A: An example of dimensional data

The data presented in sub-section 3.2 is dimensionless. As was observed in Figure 4, $\alpha$ (dimensional decay rate) found using viscoelastic models may not peak. This because $\hat{\alpha}$ versus $\hat{\lambda}_o$ curves of the viscoelastic models are normalised using open-water wavenumber. Recently, it has been shown that instrument noise and local non-linear wave generation of high-frequency waves
may cause a peak in short-wave regime which may not be real Thomson et al. (2021). To make it clearer, the comparison between field measurements documented in Wadhams et al. (1988) and predictions of models are presented in a dimensional way (Figure A1). As seen, field data reaches a peak value at wave period of $\approx 5.15\ s$, though the decay rates predicted by viscoelastic models decrease under the increase of wave period (i.e. they never peak). Interestingly, decay rate versus wave period curve predicted using the pure elastic model (RP with an additional added mass term) reaches a peak value in the
short-wave regime.

*Code availability.* The code used for construction of the curves can be provided upon reasonable request.

*Data availability.* This paper does not use any dataset. All the experimental data were extracted from Figures or Tables of other references listed in Table 1 (Voermans et al. (2021); Wadhams et al. (1988); Meylan et al. (2014); Yiew et al. (2019); Sree et al. (2018)).

*Author contributions.* ST and AVB conceptualized the research. AVB supervised the research and provided the funding. ST formulated the
models and visualized the data. ST and AVB discussed the results. ST wrote the draft of the paper. AVB reviewed the paper.

*Competing interests.* The authors declare that they have no known competing financial interests or personal relationships that could have appeared to influence the work reported in this paper.

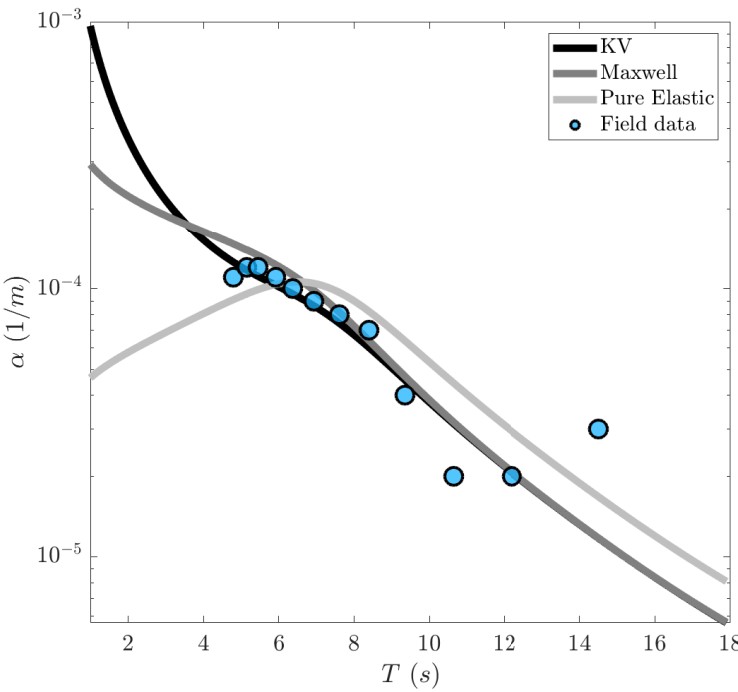

**Figure A1.** Comparison between $\alpha$ versus $T$ curves constructed using different wet beam models against those of field measurements documented in Wadhams et al. (1988). Curves are constructed using the inputs giving the best fitting, which can be seen in Legends and captions of Figure 8.

*Acknowledgements.* ST was supported by a Melbourne Research scholarship. AVB acknowledges support through the US ONR and ONRG Grant Number N62909-20-1-2080.

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
