# Peer review of "A Collection of Wet Beam Models for Wave-Ice Interaction"

_The Cryosphere, 2022_

## Author Comment (AC1)

**Response to the respected Referee 1:**

Thank you for the opportunity to give a peer review of this interesting article, "A Collection of Wet Beam Models for Wave-Ice Interaction".

*Summary:*

The article contributes to the wave-ice interaction, especially modeling the wave decay and dispersion when surface water waves propagate through an ice cover. The authors assumed the sources of wave energy dissipation from two mechanisms: one is water wave forces, and the other is the mechanical behavior of the ice layer, denoted as the fluid-based and solid-based energy damping mechanisms, respectively. They present "wet-beam" models that introduce the wave radiation term (heave direction only) in the Euler-Bernoulli beam theory and different rheologies for ice. The considered rheologies contain Kelvin Vogit (KV) model and Maxwell model and use pure elastic material as reference. Relevant dispersion relations are deduced.

The decay rates and wavenumbers are calculated using the dispersion relations with tuned rheological parameters to fit measurements from fields and lab flumes. The measurements cover landfast ice, broken ice from fields, and two lab flumes experiment with viscoelastic material and freshwater ice. The wet beam models using viscoelastic materials can agree with the measured wave decay rates in the landfast ice and broken ice fields. However, for freshwater ice, the models cannot give a well fit for decay rate and dispersion at the same time. The discrepancy is solved by introducing three-parameter viscoelastic rheologies into their dispersion relations.

The study found that the fluid-based energy damping mechanism is dominant for long waves, and the solid-based mechanism is important for short waves. The damping term in the wave radiation plays a more important role in decay rate than the added mass term. The heave added mass term can affect the wavenumber. It is also interesting to find that the equivalent Young Modulus of an SLS-type material using Maxwell approach is close to what is measured in dry tests.

The proposed idea of considering wave radiation in modeling waves propagating through ice cover will be of interest to the readership of the journal. Please see my reports below:

Dear respected Referee, we are very thankful to you for reviewing our paper and providing constructive comments to improve the manuscript. Your general comment on our paper really motivated us to further work on the manuscript and increase its quality. You will find our replies to your comments in this letter. Also, following your comments, suggestions, and queries, we have revised some parts of the manuscript, which will be visible if we are asked to upload a new version of the manuscript, though these changes are clarified in the present letter.

*General Comments:*

1. A few typos need to be corrected, which are listed in the specific comments.

All these typos will be corrected in a new version of the paper, which is not uploaded yet.

2. Do the dispersion relations Eqs. (13-15) have multiple roots features like the models mentioned in Mosig(2015)? For example, Figure 2 of Mosig (2015) shows a root distribution in the wavenumber and attenuation domain. In other words, are there multiple roots solved from Eqs. (13-15) satisfying ki>0 in this work? If so, what are the criteria for choosing the dominant root?

We are very thankful to the respected Referee for this question. That is an interesting question, and it could be much better to address it in the previous version of the manuscript. Any of presented dispersion relationships can have multiple roots as observed and discussed in Mosig (2015) and Fox and Squire (1990). The roots of dispersion relationships can be found using numerical methods, an example is presented in Section 3 of Das (2022). In the present manuscript, we have found the dominant root by using an initial guess, which was set to be equal or greater to open-water wavenumber. Following a numerical approach, the dominant root is found. In a new version of the manuscript, it will be clarified.

3. What is the reason for using different dimensionless viscosities for KV model and Maxwell model in the last row of figure 2?

Thanks for this comment and noticing this point. It would be much better to run both models with similar dynamic viscosities. Following the comment of the Respected Referee, we have corrected this problem and changed the inputs of the last row of Figure. Dimensionless viscosity of both models is now similar. Further, the respected Referee has suggested to remove the third column in one his specific comments. This has also been done by the authors. The new version of this Figure is shown below (Figure 1 of this letter). Also, the discussion (such as citation of each panel) is revised in the new version of manuscript. In the last row, two different curves for PE material are needed to be plotted as two different Elasticity numbers are considered. It can be seen that the decrease in Elasticity number may lead to increase of decay rate over the short-wave range. It is line with the results presented in Figure 9 of Mosig et al. (2015). Note that, in Figure 9 of Mosig et al (2015), the PR model (which is identical to PE material with added mass of nil), a non-realistic Shear modulus ($\approx 3.2 \times 10^7\ Pa$) is seen to give the best fitting for decay rate over short-wave range, though a realistic Shear modulus ($\approx 9.2 \times 10^2\ Pa$) is observed to under-predict the decay rate over short-wave period. The related discussion is added to a new version of manuscript, which is not uploaded yet.

[Figure]

**Figure 1:** A new version of Figure 2 of the manuscript.

3. It is unclear what value of the added mass coefficient A is used except in Figure 4 of this manuscript.

Thanks for the comment. In all the cases, $A/\rho_w h^2$ is set to be 1. It will be clarified in the new version of manuscript.

4. Is there a comparison of wavenumber corresponding to the wave decay rate comparison with Wadhams et al. (1988) and Meylan et al. (2014) in figure 6? It would be comprehensible to have such a comparison.

The authors were keen to compare the dispersion plots against any of listed experiments. But the dispersion plots (or data) of those studies are not presented/available. As such, we were not able to compare the results of present model against those of Wadhams et al. (1988) and Meylan et al. (2014). It is clarified in a new version of manuscript which will be uploaded if we are asked to do so.

5. Do you consider the wave excitation force to be another necessary potential source? Because the excitation forces, radiation forces, and static forces are the common forces that need to be considered in hydrodynamics. It could occur in low ice concentration fields of ice floes.

This is very interesting discussion. We are thankful to the respected author for mentioning this point. If we have a look at the paper presented by Newman (1994), we can see that the term $\phi\omega^2$ can represent the wave excitation force (caused by the original wave propagating under the cover) acting on the lower surface of the solid body covering the water. It will be clarified in a new version of manuscript. However, the comment of the respected Referee inspired us to discuss another potential source. When the ice floe is not integrated (such as the ice fields of Wadhams et al. (1988) and Meylan et al. (2014)), ice-ice effects, which is also known as body-body interaction, may emerge. In such a condition, presence of a neighbor floating ice floe can lead to extra added mass forces and damping forces. Apart from that, in the gap in between any two ice floes, the water surface profile may lead to an extra damping, which is introduced as artificial damping (an example is presented in Lu et al. (2010) and (2011)). In this condition, the pressure caused by the wave is formulated as

$$p = \rho\phi\omega - i\mu\phi, \tag{1}$$

where $\mu$ is the artificial damping (Lu et al. 2010). We have not considered this artificial damping in the present research, but its consideration may affect the results presented in Figure 6. If such a term is applied, different values of the viscosity may work when any of models is used. The related discussion will be added to Section 3.5 of the manuscript.

***Specific Comments:***

Line 117, Eq. (9), shear stress modulus G_E is equal to shear modulus G. Do you mean G is the elastic modulus or Young's modulus?

Thank you for the comment. $G_E$ is the dynamic shear modulus, which can include storage modulus (real component) and loss modulus (imaginary component). For a pure elastic material, the imaginary component is nil, and dynamic modulus equals shear modulus ($G$) of the material. It is now clarified in the new version manuscript.

Line 157, ko is not claimed.

Thank you for the comment. $k_o$ is the open-water wavenumber and will be introduced in the new version of manuscript.

In the bottom row of Figure 2, the Elasticity number corresponding to the dashed gray curve is not specified. By the way, the right column could be removed since the data are already presented in the other columns.

Thanks for the comment. The curves presented in the right column are also presented in the two other columns. Thus, the last column is removed the new version of manuscript, which is not uploaded yet. Also, the Elasticity number of the PE model, presented in the last row is added to Figure 2 of the new version of manuscript. Please see Figure 1 of this letter.

In figure 3, the FS model corresponding to the blue curve is not defined in the left panel. in the right panel, what is the reason for the sudden drop of the blue curve near the nondimensional wavenumber = 580.

We are very thankful to the respected Referee for this comment. The relationship for FS model will be presented in the new version of paper. In relation to sudden drop, this was a subtle point which had not been noticed by the authors. It seems to be an error of the code used for calculation of dominant root of dispersion relationship, which may happen when the fluid damping is set to be zero. The error was due to the initial guess related to long wavelength. In the previous version of manuscript, the initial guess, related to this plot, was set to be much larger than that of open-water wavenumber, which resulted to a sudden jump at dimensionless open-water wavelength ($\approx 580$). We have found it very interesting, and also clarified it in a new version of manuscript which will be uploaded if we are asked to do so. The new version of this Figure is also shown below (Figure 2 of this letter).

[Figure]

**Figure 2:** A new version of Figure 3 of the manuscript.

.

Line 230, it seems to be a typo, change the word 'travailing' to 'traveling'

Thanks to the respected Referee. This error is corrected in the new version of manuscript.

Line 243, I feel the paragraph is confusing, except "The heave added mass coefficient is seen to affect the dispersion process of waves propagating into the cover with lower Rigidity", which can be read from Figure 2(right). It is acceptable to continue with " the heave added mass coefficient can …". But I don't see why it 'matches with' large rigidity.

We agree with the respected Referee. This paragraph is re-written.

"The heave added mass can affect the dispersion process of waves propagating into cover with lower Elasticity number by increasing the wavenumber, through its influences on the cover with lower Elasticity number are more noticeable compared to cover with larger Elasticty Number".

Line 276 typo, correct the word 'viscoelastic'.

Thanks for the comment. In the new version of manuscript, the term "viscoleastic" will be changed into 'viscoelastic' in the new version of manuscript.

Figure 6's caption, a typo, move a 'by' from '... data measured by by Wadhams et al. (1988), upper row, and Meylan et al. (2014) ...'.

Thanks to the respected Referee. This will be corrected in a new version of manuscript.

The fluid damping coefficient B of red solid curves in the legends in the top row of Figure 8 is partially missed.

We are thankful to respected Referee for pointing this out. $B$ is 100 $Pa/s$ and this problem will be corrected in a new version of the manuscript. The new version of this Figure is shown below (Figure 3 of this letter).

[Figure]

**Figure 3:** A new version of Figure 8 of the manuscript.

Line 322, change "Left and right panels … Maxwell and KV materials." to "Left and right panels … KV and Maxwell materials."

It will be corrected in a new version of manuscript.

Line 455, a grammar error in "dispersion curves Maxwell model give is sensitive to dynamic viscosity"

We are thankful to the respected Referee. "give" will be changed to "gives" in a new version of manuscript.

**References:**

Das S, 2022, Flexural-gravity wave dissipation under strong compression and ocean current near blocking point, Wavesin Random and Complex Media, DOI: 10.1080/17455030.2022.2035847

Lu L, Teng B, Sun L, Chen B, 2011, Modelling of multi-bodies in close proximity under water waves—Fluid forces on floating bodies, Ocean Engineering, 1403-1416.

Lu L, Cheng L, Teng B, Sun L., 2010. Numerical simulation and comparison of potential flow and viscous fluid models in near trapping of narrow gaps. J. Hydrodyn. 22 (5s1), 120–125.

---

## Author Comment (AC2)

My apologies to the authors for getting to this review later than I anticipated when I accepted the job. The delay is especially unfortunate as there seems to be a fundamental error in the theoretical framework of the study that means I cannot recommend revisions that give a pathway to publication.

Dear respected Referee, we are very thankful to you for your comments and the time you spent reading the manuscript raising your concerns. We have noticed that you think that there is a fundamental error in formulating this research, but we believe there is a possible misunderstanding leading to such a conclusion. We have viewed your comment as an opportunity. This, in essence, helped us enhance the quality of the manuscript as we can avoid any misunderstanding when we clarify the general idea behind the model. Our responses to your comments (marked with blue colour) are listed below. Also, following your comments, we have made some changes to a new version of the manuscript, which will be visible when a new version of the manuscript would be asked to be uploaded.

The authors are proposing a model for wave propagation in ice covered water that includes wave radiation forces (added mass and heave damping), which they say are absent in most models. However, this is not correct as others (e.g. Squire, Meylan and co-workers) have developed many models that include radiation forces (none of which are referenced). Their models of elastic ice floes contain the rigid body modes of heave and pitch (in 2D) as well as elastic modes (see e.g. Meylan & Sturova, 2009, Journal of Fluids and Structures). Here, the authors have attempted to incorporate radiation forces directly into a dispersion relation for the floating ice but its implementation appears to be incorrect. Consider the damping term, which should express the transfer of energy from the body motion to radiating waves, so that no energy is lost from the wave–ice system. It should not, as it does here, induce an imaginary component of the wavenumber and hence wave energy dissipation.

We are thankful to the respected Referee for this comment and raising this concern. The models used for wave-ice interaction can be either related to water wave interaction with finite length body (or bodies) and water interaction with infinite or semi-infinite length ice cover (please see the introduction of Xu and Guyenne 2022, which introduces these two approaches as discrete-floe models and continuum models). The former approach may be used in the calculation of the wave-induced motions of any flexible body (*e.g.* ships, barges, energy converter devices, *etc*), wave scattering, wave transmission and may have applications in many different fields, ranging from oceanography to hydrodynamic of ships. The latter may be used for calculation of the ice-induced energy decay. If the ice cover is assumed to be semi-infinite, the ice edge dynamic and wave reflection can also be addressed. As known, continuum models can be used to formulate dispersion relationships (please see the introduction of Xu and Guyenne 2022), which can be utilized to calculate the wavelength (or wavenumber) and energy decay rate of waves travelling in covered water (please see Chapter 3 of Dean and Drlymple (1984)).

In the present study, as is explained in the Abstract, Introduction and Section 2, we are studying an integrated ice cover that is expanded over a long way, which means this research focuses on continuum models, not the finite length ones. Accordingly, previous studies introduced in the literature of the submitted manuscript highlight the wave interaction with infinite length or semi-infinite length covers (continuum models). The respected Referee is referred to different parts of the manuscript where the problem is introduced. We have listed some of them below:

Line 1: *"Theoretical models for the prediction of decay rate and dispersion process of gravity waves traveling into an **integrated** ice cover are introduced"*

Line 27: *"Mathematical modeling of the wave-ice interaction has first received the attentions of researchers in the 19th century. The first model was developed by Greenhill (1886), who formulated harmonic motions in a fluid domain covered with an elastic beam. To build the model, he assumed that the ice extent was spanned over **an infinite way** and the solid body had relatively small motions."*

Line 74: *"Consider a two-dimensional fluid domain containing water. **The domain extend is stretched over an infinite length**."*

To avoid any misunderstanding, we have added the following to a new version of the manuscript, which is not uploaded yet.

*"The models assuming that ice length is spanned over an infinite length are known as continuum ones. Using different assumptions made for the fluid and solid behaviour, their dispersion relationships are developed. The real and imaginary part of a dispersion relationship inform us about the wavenumber and wave decay rates. The wet beam models developed in the present research are formulated using such hypothesis (i.e. an infinite length body covers the whole fluid domain)."*

The *missing references* are related to **finite-length** problems in which the radiation problem is considered. The reason that these studies were not mentioned in the previous version of the manuscript was that they are basically built to analyze a different physical problem (i.e., a finite length one). But it is worth noting that the present research is inspired by ship hydrodynamics (finite-length problems) where the radiation force was and still is considered. We believe that we should avoid any misunderstanding. We are thankful to the respected Reviewer for mentioning this point. So, we will add a brief explanation to introduce the finite-length problems, explaining that they are mostly developed for calculation of wave transmission and body responses to water waves. To address this comment, we will briefly explain the difference between those problems and the continuum ones in the new version of the manuscript. However, we will then explain that the radiation force incorporated in these models inspired us to develop the present continuum model. This may help the readers to understand how we have formulated the new model. The following will be added to a new version of the manuscript.

*"Mathematical models for finite lengths problems have also been developed over time. They can be used for calculations of the wave-induced responses of the ice and wave transmission by the ice floes. In these models, the wave radiation forces are mostly considered, though they are not included in the infinite length problems. Many examples can be found in the literature (e.g. Meylan and Squire (1994), Kohout et al. (2007), Meylan and Sturova (2009), Montiel et al. (2012)). Note that these models can also have application in hydroelastic analysis of ships, very large floating structures, breakwaters, energy converters etc (e.g. Michele et al. 2022). The radiation force considered in most of these studies motivated us to hypothesise that this force may also emerge during the interaction of an infinite length ice floe interacting with water waves."*

The respected Referee has mentioned that we have directly implemented the heave damping term in the dispersion relationship. We are thankful to the respected Referee for opening this discussion. But such a thing has not be done. As it was explained in the manuscript, the governing equation on the

elastic beam (let's call it the solid dynamic equation) is modified to a wet beam by considering the radiation force, which is related to a wet body, and finally, the dispersion relationship is built step-by-step in the way some other researchers have done to model infinite length problem. The respected Referee is referred to line 96 of manuscript, where the dry beam equation is introduced as

*"The above equation is formulated for a dry beam (see panel a of Figure 1), which vibrates under the force of water waves"*

In the next step, the respected Referee is referred to line 104, where the wet beam equation is described as:

*"Equation 6 is an extended version of the Euler-Bernoulli beam model which is adopted for a beam exposed to water radiation forces"*.

The respected Referee has mentioned that the *heave damping is directly implemented into dispersion equation* (equations 12, 13 and 14). But, as discussed, the radiation force is incorporated in the solid dynamics equation (6) (not in the dispersion relationship), and then in the next steps the dispersion relationship is extracted by establishing the equation governing the free surface (Equation 7).

As we have mentioned earlier, we are inspired by ship hydrodynamics, and the way that motions equations of flexible floating bodies are formulated. *i.e*, Equation 6 comes from our understanding of ship hydroelasticity. We refer the respected Referee to one of the most famous references of ship hydroelasticity, Newman (1994). In this reference, solid dynamic equations related to elastic modes of a deformable body are formulated and radiation forces are also implemented (please see Equation 3.8).

The respected Referee has commented that *"Damping term, which should express the transfer of energy from the body motion to radiating waves, so that no energy is lost from the wave–ice system."*

We are thankful for this comment. But the radiation damping (called damping term by the respected Referee) can dampen the energy of the body, motions of which are coupled with those of fluid. To understand this fact, we refer this respected referee to the definition of radiation damping, introduced by McCormick (2010) (please see page 352 of McCormick (2010)):

*"As stated previously in the chapter, the body motions also result in a transfer energy to the sea, and the transferred energy flux is away from the body. This results in an energy loss to the body motion. The resulting damping effect on the body motion is called the radiation damping."*

As clearly explained, radiation damping (caused by the motion of a solid body) results in the energy loss of the body motion. For our case, the solid body is the ice cover, the motions of which result in radiation, causing the energy loss of the ice motion (waves propagating in the ice cover), which is coupled with water motion. Simply stated, a proportion of the energy of flexural-gravity waves is turned into radiation energy. If the energy of the body (which is coupled with the gravity waves) is lost, the wave-body energy (the energy of original wave) is lost, and is turned into the radiation energy, which is different from the wave-body energy (the energy of original wave). Please note that, we aim to study the loss of the energy related to the wave-body system (the energy of original wave).

Here, we need to emphasise on the fact that the term *dissipation* is just a word here. It does not necessarily refer to the conversion of wave-energy into heat. For example, if the surface waves lose mechanical energy to the surface current in the upper ocean, or to generating the winds in the atmospheric boundary layer, their energy is lost and will never come back to them. Thus, from the

point of view of modelling these surface waves, their energy is dissipated, regardless the fact that it did not turn into the heat, but rather into other form of mechanical energy (for water currents and for wind). In any model of these surface waves, this will result in imaginary part of their wavenumber. The same happens here. The original gravity wave which interacts with the ice cover, can be dissipated (imaginary wavenumber, i.e. decay of their mechanical energy), while the other mechanical energy (related to radiated waves) grows.

We believe to avoid such a misunderstanding, we can refer the readers to the textbook of *Ocean Engineering Mechanics with its Applications* (McCormick (2010)) and explain the above reasoning in the manuscript.

*"Note that the radiation damping can cause the transfer of energy to the water, resulting in energy loss of the ice motion. This consequences in the loss of the energy of the wave-ice system. Readers who are interested in the radiation damping are referred to Chapter 10 of McCormick (2010)."*

Finally, the general idea is based on the wave-induced motions, coming from textbook of *Marine Hydrodynamics.* We assume that the there is an exciting force caused by the progressive wave (please see equation 1 of the present letter), and a force related the radiation of waves (please see equation 2) due to oscillations of the body (which is the ice layer). The respected Referee is referred to Newman (1977)**,** Newman (1994)**,** Bishop and Price (1974, 1976)**.**

We assume that there is no rigid body motion since the ice cover is very long and waves are much shorter, compared to the ice length. Wave motion emerges in the body, since the problem is 2D and we have used the Euler-Bernoulli beam theory, no rotational motion emerges, we only consider modes related to vertical bending as the beam length is spanned over a long way. The beam has an infinite length, so the elastic mode(s) of vibration are what are found through the dispersion relationship. This is exactly like what has been done by other researchers studying semi-infinite or infinite length beams vibrating under any exciting force (see an example in Saito and Murkami (1969)).

We have two different types of force acting on the solid ice beam. These forces are independent, but their combination helps us formulate the dynamic motion of the upper layer and thus build the dispersion relationship. The first force is exciting one caused by the progressive (let's call it wave-induced force) waves is found through

$$f = -i\omega \int \phi \boldsymbol{h} dS \tag{1}$$

The respected reviewer is referred to Senjanovi et al. (2009) (also check Chapter 6 of Newman 1977).

The fluid force related to radiation force can be formulated as

$$f_R = -\omega^2 a - i\omega b \tag{3}$$

The above equation is well accepted by the ocean engineering community, studying radiation force (researchers and engineers aiming to study motions of ships and any floating bodies). The imaginary component of the radiation force is the damping term and has a theortical background (the respected Referee is referred to Newman (1977), Faltinsen (1993), Newman (1994), McCormick (2010), Senjanovi et al. (2009)). In all these references (which are either textbooks or journal papers), the damping force caused by the radiation is the imaginary component of the force. Note that damping force is in phase with the rate of displacement of the solid motion, and the heave damping is in phase with the vertical acceleration. The work done by the added mass force (over a cycle) is zero, though

that of damping is non-zero, and is responsible for the loss of the energy (the respected Referee is referred to the Chapter 3 of Faltinsen (1993)). To avoid any misunderstanding, we have added the following to the manuscript:

"*Note that to formulate the wet beam equation, two different forces are assumed to act on the solid body: radiation and the wave-induced force. The work done by the heave added mass is in the phase with acceleration, and the work done by heave damping is in phase with vertical velocity of the solid body, the work of which is responsible for the loss of the energy of wave propagating in the ice cover. Since the cover has assumed to have an infinite length, rigid modes are not involved, and the only elastic mode that involves is the one that is found through finding the root of the dispersion relationship.* "

The term in the dispersion relation used to represent heave radiation is identical to that derived from the Robinson–Palmer model, which has been used by many previous authors and shown to be capable of giving reasonable predictions of wave attenuation (again, lots of references missing). Therefore, key findings, such as "decay rates were observed to be poorly predicted if the fluid-based energy damping is not taken into account", must be reinterpreted in the context of the RP model and lose their novelty.

We are thankful to the respected referee for this comment. Again, this helps us avoid any misunderstanding. The authors did not claim that the term representing heave radiation is mathematically different from what was introduced by Robinson–Palmer. What the authors have insisted on in the manuscript is that the radiation force, including *heave added mass* and *heave damping,* can be incorporated when the dispersion relationship of waves propagating into any body (either elastic or viscoelastic) is formulated. The fact is that, in the previous studies, the fluid-based damping (or let's call it the term $-iB\omega$) and the solid-based damping have not been considered simultaneously (the respected referee is referred to the sub-section 2.5 of Squire et al. 2021). In the present research, however, both two dissipative mechanisms, *heave damping* (fluid- based, $-iB\omega$) and the *viscoelastic effects* (solid-based) are assumed to emerge at the same time.  Also, we would like to also explain that, in a more general view, the $iB\omega$ term can represent a combination of the viscous effects (let's call it friction) and heave radiation damping, and any other linear damping. As different linear damping terms may emerge at the same time (please see Section 10.1 of McCormick (2010).

The respected Referee has mentioned that the PR model has been used by many other researchers, but the fact is that the PR model is developed for an elastic plate (*i.e.* material is not assumed to be viscoelastic, the sub-section 2.5 of Squire et al. 2021 describes this) and they have never considered the *viscoelastic effects* when they use any $iB\omega$ term.  Simply stated, other researchers have never considered *fluid-based* and *solid-based* damping at the same time, while we have tried to do so in the present research, and we believe this is the contribution of the present study. In the introduction section of the manuscript, we have clearly explained our hypothesis, which makes our research different from any previous (line 60 of manuscript).

"*The coexistence of solid-based and fluid-based dissipation mechanisms in formulations may help us reconstruct the decay rates a more realistic condition*"

The dispersion equation of an elastic beam we presented (Equation 12) is related to the case we only consider fluid-based energy damping, and no solid-based energy damping is considered. This is very

similar to PR model. We have honestly and openly mentioned this fact (please see line 133 of manuscript):

*"Another version of this relationship has been previously documented by Mosig et al. (2015), which is originally formulated by Robinson and Palmer (1990) modeled interaction of an elastic body interacting with water waves. But the added mass term is included in the above equation, which makes it different from what was presented by Mosig et al. (2015)."*

We would like to ask the respected Referee to consider the two other dispersion relationships (Equations (13) and (14)), both of which are developed by considering fluid-based (heave damping) and solid-based (loss modulus related to viscoelastic martial) energy damping (they are built based on our hypothesis). We need to also recall that two SLS models are also introduced in the paper (Equations 26 and 27), which are completely different form PR model (as they use three-component solid models). These are not apparently considered by the respected Referee. We would like to ask the respected Referee to consider these dispersion relationships as well.

The respected Referee has commented that the key findings *"decay rates were observed to be poorly predicted if the fluid-based energy damping is not taken into account"* should be reinterpreted the context of the RP model. This quote from the Conclusion Section was mentioned without consideration of the whole paragraph. We would like to ask the respected Referee to consider the previous sentences as well. The complete paragraph is (please see line 465):

*"Predictions of models and field measurements were quantitatively compared against each other. **KV and Maxwell** models were seen to reconstruct the decay rates and dispersion process of landfast ice with a great level of accuracy. Decay rates were observed to be poorly predicted if the fluid-based energy damping is not taken into account, suggesting that this mechanism has a very important role in ice-induced energy decay over the range of long waves"*

It is clear we talk about **KV and Maxwell models**, not the elastic one (which is the only one that is similar to PR model). When we say, *"decay rates are poorly predicted when the fluid-based energy damping is not considered"*, we are making a conclusion related to the behaviour of models built for KV and Maxwell materials, which also include *solid-based energy* damping. We need to recall that these two dispersion relationships are different from PR model as the solid-based energy damping is included. To make it clearer, a PR model uses a *one parameter* solid model, but KV and Maxwell models use *two parameter* ones. In general, **the key finding of the research is that the solid-based and the fluid-based energy damping can be considered at the same time as there is no inconsistency between fluid damping and solid damping**. We again refer the respected Referee to the hypothesis of this research. What the respected Referee has mentioned is related to an elastic beam, not the two viscoelastic ones and the related findings are documented by Moisg et al. (2015) or previous researchers (found through using the PR model) are related to an elastic plate. We have mentioned this in the manuscript.

The respected Referee has pointed that many references are missing. When we wrote the early version of the manuscript, we only referred the readers to two main studies introducing the PR model are presented in the paper.  The original paper of Palm and Robinson (1990) and the paper authored by Mosig et al. (2015). We agree with the respected Referee that it would be better to refer to some other studies which have used the PR model. But we need to emphasise on the fact that all these references present dispersion relationship of an elastic ice (with inclusion of the term $-iB\omega$) and no viscoelastic behaviour is considered as mentioned above. For instance, the respected Referee is referred to Appendix A of Williams et al. (2013a) (Here the appendix is called *Thin **elastic** plate model with the inclusion of damping*). But to avoid any misunderstanding, we will add these references

(Squire and Fox (1992), Williams et al. (2013a, b), Squire et al. (2009)) to Introduction section of the paper and clarify that dispersion relationship presented in these papers are different from what is presented in Equation (13), (14), (26) and (27), and only consider $-iB\omega$ term. Please again note that the hypothesis of the present research is that terms $-iB\omega$ and $-i\frac{G''\omega^2}{h^3}$ can be considered at the same time.

In total, we agree with the respected Reviewer that the references should be included in the manuscript as we can provide a better understating of what has done before. We have addressed the comment of the respected Referee in the new version of manuscript.

"Note that the previous version of the dispersion relationship of pure elastic material was presented in some other studies (e.g., Williams et al. (2013a, b), Squire et al. (2009)), and is mostly called the RP model. But note that this model has only considers one dissipative mechanism, though in this research, it is aimed to show that coexistence of two different dissipative mechanism (fluid-based and solid-based) help us predict decay rate and dispersion relationship."

Apart from that, other researchers, specifically, Mosig et al. (2015) have shown that the PR model may provide us with a perfect fitting with experimental data. But, in Mosig et al. (2015), elastic modulus of $3.2 \times 10^7$ Pa was observed to give the best fitting, though the shear modulus of ice is expected to be greater than 1 GPa. This should be explained through investigating the difference between nature of the viscoelastic models and the PE one, which gives different behaviour for $\alpha_i$ vs. $\omega$. The log-log plots of decay rates as a function of frequency are presented. This again provide clear evidence of the novelty of the present research. This is a clear piece of evidence showing that the presented decay rates of dispersion relationships of viscoelastic materials (Equations (13) and (14)) show different behavior as compared against those of the PR model in large frequencies.

Back to the problem with non-realistic inputs for elastic modulus that may give perfect matching for decay rate when a pure elastic model is embarked. As seen in Figure 1 (of this letter), a PE model gives a dependency of $\omega^3$ up to a specific limit, and then it gives a dependency of $\omega^{-0.35}$ at large frequencies (short wave periods). If the elastic modulus of the material is decreased, that specific limit is shifted towards larger frequencies (please see left column of Figure 2 of the paper). This is the main reason that in sub-section 7.2 of Mosig et al. (2015), the curve predicted by a realistic elastic modulus diverged from the field data at large frequencies (short periods), but the one predicted by a low elastic modulus followed the data. However, viscoelastic models do not show such behavior as $\alpha_i \propto \omega^{0.5}$ at large frequencies.

This is very interesting point. The authors believe they need to mention this point in the new version of manuscript (not uploaded yet) to avoid any misunderstanding and show the difference between the viscoelastic models and Equation (12) (which is also similar to PR model). The following explanting and Figure have been added to that eversion of manuscript.

"One of the main differences between the PE model and the viscoelastic ones is the dependency they give for decay rate as a function of wave frequency. The log-log plots of $\alpha_i$ vs. $\omega$ are plotted in Figure 4 to show these differences. The PE model gives a dependency of $\omega^3$ up to a specific limit, and it gives

a dependency of $\omega^{-0.35}$ at large frequencies. The two other models give the same dependency the PE model gives at short frequency. But at large frequencies, these models give a dependency of $\approx \omega^{0.5}$. This well shows that when viscoelastic models are used, $\alpha_i$ as a function of $\omega$ shows a different behavior, especially at larger frequencies. With the decrease in elasticity modulus of the material, the range over which $\alpha_i$ follows $\omega^3$ increases, which means that the viscoelastic effects become less dominant. This may be one of the main reasons that we may be able to fit the experimental or field data when the PE model is embarked if we remarkably decrease the elastic modulus of the material (please see sub-section 7.2 and Figure 9 of Mosig et al.). But when a viscoelastic model with fluid damping is used, a more realistic elastic modulus works."

[Figure]

Figure 1. Log-log plots of decay rates as a function of wave frequency. (This is Figure 4 in the new version of the manuscript). The Young's Modulus is set to be 3 Gpa and ice thickness is set to be 35 CM.

Aside from the issues with the radiation force, the paper comes across as contributing yet more models of waves in ice covered waters with parameters tuned to particular datasets but without the

general predictive capabilities needed for improved understanding of the wave–ice system. It is not surprising that adding more tuning parameters allows for better agreement with observations. Advances require connections between the parameter values and the ice properties associated to the different datasets.

We are thankful for this comment. The aim of the present research as explained in the manuscript is to provide models that include both fluid-based and solid-based energy damping, which helps us predict wave decay rate and wave dispersion with less limitation. So, we need to test how we can fit models with the field or flume data. Note that, doing so can be viewed as a common task when a new model is introduced. We refer the respected Referee to two recent references (Xu and Guyenne 2022 and Southerland et al. (2019)). In these two studies, new models for prediction of the ice-induced attenuation are developed, and the accuracy level of models in the prediction of the decay rate is assessed by tuning up inputs. Similar approach is followed in the present research.

The respected Referee has commented "*Advances require connections between the parameter values and the ice properties associated to the different datasets*". In response to what the respected Referee has mentioned, we need to clarify that this manuscript is not limited to tuning up the parameters and developing model. In the manuscript, we have included a section in which we entirely discuss our understanding of the models and why different inputs give the best fitting. Following points were explained:

1- Both fluid-based and solid-based energy damping may emerge when water waves propagate into ice. The respected referee is referred to line (410):

*"Energy dissipation is caused by solid-based energy damping and fluid-based energy damping. The former is dominant over high frequencies (corresponding to short waves in an open-water condition), and the latter is dominant over small frequencies (corresponding to long waves in an open-water condition)"*

2- The effect of solid-based (the effect of viscoelasticity of the material) damping may be dominated over a wider range of frequency when the elastic modulus of a material is increased. This had been shown in Figure 2 (a new version of this Figure is provided). The respected referee is referred to line (414):

*"With an increase in Elasticity number, the wavelength range over which solid-based energy damping is dominant becomes wider"*

3- It is discussed that KV and Maxwell models can predict the decay rates of the landfast ice if the fluid-based energy damping is activated. We need to emphasize that it is different from the PR model as this model does not consider any viscoelasticity. It is explained that the viscosities that gives best fitting are different, but we believe the KV material is a better representative as a landfast ice as it is likely to be more similar to a solid viscoelastic material (i.e. the ice is a KV material). The respected referee is referred to line 423:

*"Both viscoelastic models were seen to construct the decay curves, fairly follow field measurements. But the dynamic viscosity values were seen to be much different. A KV material may be a more realistic indicator of ice behavior as the lanfast ice is expected to be solid."*

4- We then explain that, for the broken ice field (Figure 6), viscoelastic-based models give fitting with different viscosities as compared against viscosities by applying which the same model gave the best fitting for lanfast ice. We have presented a discussion on this difference. We have explained that a volume fraction method may help us understand this. If we assume the ice layer is a mixture of water and ice, then the viscosity of the layer can be represented by a volume fraction method. It is explained that for a KV material, the dynamic viscosity is lower as compared against that of landfast ice, and for the Maxwell material, the dynamic viscosity is lower as compared against landfast ice. It means that if water entrapped in between broken ice floes increases the viscosity of the whole layer, the Maxwell material can be describing the ice behavior, otherwise KV material is a better representative of the ice behavior. This is also seems to be a potential discussion on "*connections between the parameter values and the ice properties*". The respected referee is referred to line 433:

"*The KV model gives the best fitting of decay rate when the dynamic viscosity is reduced, compared to the landfast ice. In contrast, the Maxwell model gives the best fitting with a larger dynamic viscosity. Thus, what can be concluded is that if water reduces the dynamic viscosity of the upper layer, the KV model will be more likely to be a better indicator of the ice behavior. Otherwise, the Maxwell model prescribes the mechanical behaviour of the material.* "

5- We have shown that for the freshwater ice formed in the flume viscoelastic models cannot construct relative wavenumber versus open-water wavelength curve and decay rates versus open-water wavelength with the same inputs. This motivated us to build dispersion relationships by assuming SLS models and using three-parameters viscoelastic models. When we found the parameters giving the best fit for both relative wavenumbers and decay rates, we observed that the SLS Maxwell model works with a realistic equivalent elastic modulus (3Gpa), though the SLS KV model works with a non-realistic value (2.5 $10^4$ Pa). This again has provided us with understandings of "*connections between the parameter values and the ice properties* ". We concluded that the freshwater ice, which was also very young (around 8 hours old), may not be properly modelled by two-parameter viscoelastic models (Equations (13) and (14)), though the SLS Maxwell model provide us with accurate outputs with reasonable inputs. Please see Line 440 of the manuscript.

*"Models, however, were not able to capture the dispersion process under freshwater ice covers with the same input observed to give the best decay rates. Effective values were seen to construct dispersion plots with an acceptable level of accuracy. This has been observed and reported by other scholars over the last two decades, who measured the wavelength and phase speed of disintegrated elastic/viscoelastic covers. The interesting point is that, when a Maxwell model is used, the dynamic viscosity can affect the dispersion process. This motivates us to build other models which formulate the storage modulus by applying the dynamic viscosity. Two available linear models were introduced. One is SLS KV and the other is SLS Maxwell. Both models include two springs and one dashpot. The equivalent Young Modulus giving the best fitting when SLS KV is used is 2.5× $10^4$ Pa, which is much smaller compared to*

*that of real freshwater ice. But, for SLS Maxwell, the equivalent Young Modulus was seen to be $3.3\times 10^9$ Pa which is close to that of freshwater ice."*

In the end, we are again very thankful to the respected Referee for her(his) comments and discussions. As his(her) concern helped us provide more details about the model and avoid any misunderstanding.

**References**

Bishop RED, Price WG, On the relationship between dry modes and wet modes in the theory of ship response, Journal of Sound and Vibration, 45(2), 157-164.

Bishop RED, Price WG, On model analysis of ship strength, 45(2), 157-164.

Dean RG, Dalrymple RA, 1991, Water Wave Mechanics for Engineers and Scientists, Advanced Series on Ocean Engineering: Volume 2, https://doi.org/10.1142/1232

Faltinsen OM 1993, Sea loads on ships and offshore structures, Cambridge University Press

Kohout AL, Meylan MH, Sakai S, Hanai K, Leman P, Brossard D, 2007, Linear water wave propagation through multiple floating elastic plates of variable properties, Journal of Fluids and Structures, 23(4), 649-663.

McCormick ME, 2010, Ocean Engineering Mechanics with Applications, Cambridge University Press.

Meylan MH, Sturova IV, 2009, Time-dependent motion of a two-dimensional floating elastic plate, Journal of Fluids and Structures, 25(3), 445-460.

Montiel, F., Bennetts, L., Squire, V., Bonnefoy, F., & Ferrant, P. (2013). Hydroelastic response of floating elastic discs to regular waves. Part 2. Modal analysis. *Journal of Fluid Mechanics, 723*, 629-652. doi:10.1017/jfm.2013.124

Mosig JEM, Montiel F, Squire VA, 2015, Comparison of viscoelastic-type models for ocean wave attenuation in ice-covered seas, J. Geophys. Res. Oceans, 120, 6072–6090, doi:10.1002/2015JC010881.

Newman JN, 1977, Marine Hydrodynamic, MIT Press

Newman JN, 1994, Wave effects on deformable bodies, Applied Ocean Research, 16, 47-59.

Saito H, Murakami T, 1967, Vibrations of an infinite beam on elastic foundation, Bulletin of JSME, 534, 200-205.

Senjanvoic I, Melanica S, Tomasevic S, 2009, Investigation of ship hydroelasticity, Ocean Engineering, 35, 523-535.

Squire, V. A., Vaughan, G. L., & Bennetts, L. G. (2009). Ocean surface wave evolvement in the Arctic Basin. Geophysical Research Letters, 36, L22502. https://doi.org/10.1029/2009GL040676

Squire VA, Kovalev PD, Kovalev DP, 2021, Resonance and interactions of infragravity waves with sea ice, Cold Regions Science and Technology, 182, 103217.

Sutherland G, Rabault J, Christensen KH, Jensen A, (2019), A two layer model for wave dissipation in sea ice, Applied Ocean Research, 88, 111-118.

Williams TD, Bennetts LG, Squire VA, Dumont D, Bertino L, Wave–ice interactions in the marginal ice zone. Part 1: Theoretical foundations, Ocean Modelling, 71, 81-91.

Williams TD, Bennetts LG, Squire VA, Dumont D, Bertino L, Wave–ice interactions in the marginal ice zone. Part 2: Numerical implementation and sensitive studies along 1D transects of ocean surfaces, Ocean Modelling, 71, 92-101.

Xu B, and Guyenne P, 2022, Assessment of a porous viscoelastic model for wave attenuation in ice-covered seas, Applied Ocean Research, 122, 2022, 103122.

---

## Author Response (AR1)

Dear Editor and dear Reviewers,

We would like to take this opportunity to express our appreciation for your valuable and constructive comments. We have revised the manuscript following the comments and suggestions of respected Editor and Reviewers. All revised parts in the been highlighted with red color in the manuscript. Also, we have provided point-to-point responses to the comments of both Reviewers, following in blue colour.

**Modifications based on the suggestions of the respected Editor**

Dear respected Editor, we are very thankful to you for reviewing the manuscript, and providing us with your comments on Originality, Scientific Quality, Significance, and Presentation quality of the manuscript. Following your comments and suggestions on Originality and Significance of the manuscript, we have done two modifications, outlined below:

1 – We have added clear statements regarding what new approach and what new insight are presented in the paper (line 88 of the new version of manuscript).

2 – We have explained the possible application of the research in a more explicitly environmental context in Abstract (line 16 of the new version of manuscript), Introduction (lines 24 – 30 of the new version of manuscript) and Conclusion (line 560 of the new version of manuscript).

**Point by point response to Reviewer 1**

Thank you for the opportunity to give a peer review of this interesting article, "A Collection of Wet Beam Models for Wave-Ice Interaction".

*Summary:*

The article contributes to the wave-ice interaction, especially modeling the wave decay and dispersion when surface water waves propagate through an ice cover. The authors assumed the sources of wave energy dissipation from two mechanisms: one is water wave forces, and the other is the mechanical behavior of the ice layer, denoted as the fluid-based and solid-based energy damping mechanisms, respectively. They present "wet-beam" models that introduce the wave radiation term (heave direction only) in the Euler-Bernoulli beam theory and different rheologies for ice. The considered rheologies contain Kelvin Vogit (KV) model and Maxwell model and use pure elastic material as reference. Relevant dispersion relations are deduced.

The decay rates and wavenumbers are calculated using the dispersion relations with tuned rheological parameters to fit measurements from fields and lab flumes. The measurements cover landfast ice, broken ice from fields, and two lab flumes experiment with viscoelastic material and freshwater ice. The wet beam models using viscoelastic materials can agree with the measured wave decay rates in the landfast ice and broken ice fields. However, for freshwater ice, the models cannot give a well fit for decay rate and dispersion at the same time. The discrepancy is solved by introducing three-parameter viscoelastic rheologies into their dispersion relations.

The study found that the fluid-based energy damping mechanism is dominant for long waves, and the solid-based mechanism is important for short waves. The damping term in the wave radiation plays a more important role in decay rate than the added mass term. The heave added mass term can affect the wavenumber. It is also interesting to find that the equivalent Young Modulus of an SLS-type material using Maxwell approach is close to what is measured in dry tests.

The proposed idea of considering wave radiation in modeling waves propagating through ice cover will be of interest to the readership of the journal. Please see my reports below:

Dear respected Referee, we are very thankful to you for reviewing our paper and providing constructive comments to improve the manuscript. Your general comment on our paper really motivated us to further work on the manuscript and increase its quality. You will find our replies to your comments in this letter. Also, following your comments, suggestions, and queries, we have revised some parts of the manuscript. Please note that, after the interactive discussion we decided to remove the justification of the consideration of the forces through the radiation problem, as it can be physically dubious.

*General Comments:*

1. A few typos need to be corrected, which are listed in the specific comments.

All these typos will be corrected in a new version of the manuscript.

2. Do the dispersion relations Eqs. (13-15) have multiple roots features like the models mentioned in Mosig (2015)? For example, Figure 2 of Mosig (2015) shows a root distribution in the wavenumber and attenuation domain. In other words, are there multiple roots solved from Eqs. (13-15) satisfying $k_i>0$ in this work? If so, what are the criteria for choosing the dominant root?

We are very thankful to the respected Referee for this question. That is an interesting question, and it could be much better to address it in the previous version of the manuscript. Any of presented dispersion relationships can have multiple roots as observed and discussed in Mosig (2015) and Fox and Squire (1990). The roots of dispersion relationships can be found using numerical methods, an example is presented in Section 3 of Das (2022). In the present manuscript, we have found the dominant root by using an initial guess, which was set to be equal or greater to open-water wavenumber. Following a numerical approach, the dominant root is found. It is clarified in the new version of manuscript (line 180 of the new version of manuscript).

3. What is the reason for using different dimensionless viscosities for KV model and Maxwell model in the last row of figure 2?

Thanks for this comment and noticing this point. It would be much better to run both models with similar dynamic viscosities. Following the comment of the Respected Referee, we have corrected this problem and changed the inputs of the last row of Figure. Dimensionless viscosity of both models is now similar. Further, the respected Referee has suggested to remove the third column in one his specific comments. This has also been done by the authors (please see the new version of Figure 2).

4. It is unclear what value of the added mass coefficient A is used except in Figure 4 of this manuscript.

Thanks for the comment. In all the cases, $A/\rho_w h^2$ is set to be 1. It is clarified in the new version of manuscript (line 318 of the new version of manuscript).

5. Is there a comparison of wavenumber corresponding to the wave decay rate comparison with Wadhams et al. (1988) and Meylan et al. (2014) in figure 6? It would be comprehensible to have such a comparison.

The authors were keen to compare the dispersion plots against any of listed experiments. But the dispersion plots (or data) of those studies are not presented/available. As such, we were not able to compare the results of present model against those of Wadhams et al. (1988) and Meylan et al. (2014). It is clarified in the new version of manuscript (line 403 of the new version of manuscript).

6. Do you consider the wave excitation force to be another necessary potential source? Because the excitation forces, radiation forces, and static forces are the common forces that need to be considered in hydrodynamics. It could occur in low ice concentration fields of ice floes.

This is very interesting discussion. But, the wave force acting on the structure is the dynamic pressure, known as the Froude Kyrolve force. But in a low concentration field, the gap effects and the body-body interaction may happen. We have found it very interesting and have added the related explanations to the manuscript (line 128 of the new version of manuscript).

***Specific Comments:***

Line 117, Eq. (9), shear stress modulus G_E is equal to shear modulus G. Do you mean G is the elastic modulus or Young's modulus?

Thank you for the comment. $G_E$ is the dynamic shear modulus and G is shear modulus (it is clarified in line 147). Dynamic shear modulus can include storage modulus (real component) and loss modulus (imaginary component). This point is clarified in the manuscript (Line 140 of new version of manuscript). For a pure elastic material, the imaginary component is nil, and dynamic modulus equals shear modulus ($G$) of the material. It is now clarified in the new version manuscript (line 147 of the new version of manuscript).

Line 157, ko is not claimed.

Thank you for the comment. $k_o$ is the open-water wavenumber and is  introduced in the new version of manuscript (Line 198 of the new version of manuscript).

In the bottom row of Figure 2, the Elasticity number corresponding to the dashed gray curve is not specified. By the way, the right column could be removed since the data are already presented in the other columns.

Thanks for the comment. The curves presented in the right column are also presented in the two other columns. Thus, the last column is removed the new version of manuscript.

In figure 3, the FS model corresponding to the blue curve is not defined in the left panel. in the right panel, what is the reason for the sudden drop of the blue curve near the nondimensional wavenumber = 580.

We are very thankful to the respected Referee for this comment. The relationship for FS model will be presented in the new version of paper (Line 175). In relation to sudden drop, this was a subtle point which had not been noticed by the authors. It seems to be an error of the code used for calculation of dominant root of dispersion relationship, which may happen when the fluid damping is set to be zero. The error was due to the initial guess related to long wavelength. In the previous version of manuscript, the initial guess, related to this plot, was set to be much larger than that of open-water wavenumber, which resulted to a sudden jump at dimensionless open-water wavelength ($\approx 580$). We have found it very interesting, and also clarified it in a new version of manuscript (line 183 of the new version of manuscript).

Line 230, it seems to be a typo, change the word 'travailing' to 'traveling'

Thanks to the respected Referee. This error is corrected in the new version of manuscript.

Line 243, I feel the paragraph is confusing, except "The heave added mass coefficient is seen to affect the dispersion process of waves propagating into the cover with lower Rigidity", which can be read from Figure 2(right). It is acceptable to continue with " the heave added mass coefficient can …". But I don't see why it 'matches with' large rigidity.

We agree with the respected Referee. This paragraph is re-written (Line 306 of the new version of manuscript).

Line 276 typo, correct the word 'viscoelastic'.

Thanks for the comment. In the new version of manuscript, the term "viscoleastic" is changed into 'viscoelastic' in the new version of manuscript.

Figure 6's caption, a typo, move a 'by' from '... data measured by by Wadhams et al. (1988), upper row, and Meylan et al. (2014) …'.

Thanks to the respected Referee. This is corrected in a new version of manuscript.

The fluid damping coefficient B of red solid curves in the legends in the top row of Figure 8 is partially missed.

We are thankful to respected Referee for pointing this out. $B$ is 100 $Pa/s$ and this problem is corrected in a new version of the manuscript (please see Figure 9 of the new version of manuscript).

Line 322, change "Left and right panels … Maxwell and KV materials." to "Left and right panels … KV and Maxwell materials."

It is corrected in the new version of manuscript.

**Point by point response to Reviewer 2**

My apologies to the authors for getting to this review later than I anticipated when I accepted the job. The delay is especially unfortunate as there seems to be a fundamental error in the theoretical framework of the study that means I cannot recommend revisions that give a pathway to publication.

Dear respected Reviewer, we are once again thankful to you for your general comment and time you spent reviewing the manuscript. Your comment and the interactive discussion benefited the manuscript. In relation to the theortical error, we believe that there is a misunderstanding.

The authors are proposing a model for wave propagation in ice covered water that includes wave radiation forces (added mass and heave damping), which they say are absent in most models. However, this is not correct as others (e.g. Squire, Meylan and co-workers) have developed many models that include radiation forces (none of which are referenced). Their models of elastic ice floes contain the rigid body modes of heave and pitch (in 2D) as well as elastic modes (see e.g. Meylan & Sturova, 2009, Journal of Fluids and Structures). Here, the authors have attempted to incorporate radiation forces directly into a dispersion relation for the floating ice but its implementation appears to be incorrect. Consider the damping term, which should express the transfer of energy from the body motion to radiating waves, so that no energy is lost from the wave–ice system. It should not, as it does here, induce an imaginary component of the wavenumber and hence wave energy dissipation.

We are thankful to the respected Reviewer for this comment. As was discussed during the interactive discussion, in this research we aim to develop a continuum model. We are trying to develop dispersion relationship under an integrated ice cover spanned over an infinite length. To avoid any misunderstanding, it is now directly mentioned in the new version of manuscript (line 2 of new version of manuscript) and the last paragraph of introduction (line 82 of new version of manuscript).

Following this, we have also modified the introduction section by introducing finite length models (lines 43-47 of new version of manuscript) and continuum models (lines 48-57 of new version of manuscript). This provides a better picture of the common approaches used to establish models. Following this, we have provided our understanding of the limitations of the continuum models and the opportunities (this opportunity is to combine fluid and solid forces to formulate the dispersion relationship) for their improvement in lines 60-83 of the new version of manuscript.

We believe the above modifications will address the missing references related to finite length problems and will provide an idea of our motivation in this paper (the way we have viewed the limitations and opportunities).

In relation to the problem with the radiation problem, after the interactive discussion, we have decided to remove the justification as it may make understanding of the problem very hard and also may be physically dubious. We have tried to introduce the fluid forces just in the way other researchers do (line 124 of the new version of manuscript).

The term in the dispersion relation used to represent heave radiation is identical to that derived from the Robinson–Palmer model, which has been used by many previous authors and shown to be capable of giving reasonable predictions of wave attenuation (again, lots of references missing). Therefore, key findings, such as "decay rates were observed to be poorly predicted if the fluid-based energy damping is not taken into account", must be reinterpreted in the context of the RP model and lose their novelty.

We are thankful to the respected Reviewer for this comment. We believe this is a misunderstanding. The conclusion we made is related to viscoelastic models presented in this paper, not the elastic model. We believe we may avoid this misunderstanding by clearly mentioning what is new in this manuscript, which was also suggested by the respected Editor (Line 88 of manuscript). We have also mentioned that the pure elastic model is RP model with an additional term, and we are only attempting to see how it works in prediction of the decay rate, as compared to viscoelastic models (Lines 163-174 of the new version of manuscript).

Aside from the issues with the radiation force, the paper comes across as contributing yet more models of waves in ice covered waters with parameters tuned to particular datasets but without the general predictive capabilities needed for improved understanding of the wave–ice system. It is not surprising that adding more tuning parameters allows for better agreement with observations. Advances require connections between the parameter values and the ice properties associated to the different datasets.

We are thankful to the respected Reviewer for this comment. We have introduced new dispersion relationships in this manuscript, and we believe it is common practice to check the validity of the models with tuning parameters. Meanwhile, in sub-sections 3.4 and 3.5 we have tried to discuss why different inputs give the best agreement with experiments. Providing more discussion related to the mechanical behavior of the ice is out of the scope of the present research, and may need laboratory tests.

Theoretical model
* * *
The theoretical model is

- purely elastic ice, or damping in the ice from the imaginary part of the Young's modulus. The specific formulation for the damping comes either from the Kelvin-Voigt or the Maxwell rheology and gives different frequency dependance in the damping coefficient.

- damping in the fluid from B, the radiation damping coefficient. (This is the same as the Robinson-Palmer                                                                                                    model.)

- extra inertia from A, the added mass coefficient

The main novelty to me are the different ice rheologies, but the fluid damping effectively has little novelty (with the exception of A) but only introduces a more complicated (and physically more dubious) justification for the Robinson-Palmer (RP) model. I would remove the physical justification completely as (a) unnecessary and (b) physically dubious. (Note I am not proposing to remove the RP model itself as applying an old model to new data can still be interesting.)

I say it is physically dubious as the added mass and damping are usually derived from solving the hydrodynamic equations (Laplace's equation + sea floor condition + boundary condition (7)) with A=B=0 when a body is forced to oscillate. So to put them into (7) seems a bit circular. (Incidentally, in equations 5 and 6, $z^4$ should be $z_{xxxx}$.)

In the authors' reply to Reviewer 2, they talk about continuum media (I guess effective media). Maybe they are trying to represent the attenuation due to scattering by a large number of scatterers. Phase-resolving scattering models do predict that wave energy does decay into ice, but they also conserve energy. While they would not be the first authors to represent the attenuation due to scattering with a dissipative model (eg Williams, Bouillon & Rampal, 2017, The Cryosphere)(for lack of a good alternative), they aim to represent it entirely with Robinson-Palmer dissipation instead of empirically, as most authors do.

It should also be noted that scattering models give quite different results to Robinson-Palmer especially at long periods, and since Robinson-Palmer (combined with the dissipation inside the ice itself) gives realistic results in these case it begs the question of why they are bringing in scattering at all.

The authors are very thankful to the respected Reviewer for his/her comment on the theortical models presented in the manuscript. This comment was very constructive and helped authors

improve the quality of the presentation of the model and avoid any misunderstanding in the paper.

As the respected Reviewer has mentioned, it has been aimed to add the fluid damping term into dispersion relationship (combine the RP model with decay into the ice). We agree with the respected Reviewer that the Robinson and Palm (1990) has introduced fluid damping in their model (RP), which has been used by many scholars over years. In the present research, we tried to include the fluid damping into dispersion relationships of viscoelastic ice beam as we hypothesized that fluid damping term and a complex term in flexural rigidity can be considered at the same time. The first dispersion relationship (Equation 13) is therefore the RP model with an additional added mass term (which may weakly affect the wavelength), as the respected Reviewer has mentioned. We have clarified this point in the new version of manuscript that we do not view this dispersion relationship as a new one here, and the aim is to build relationships for viscoelastic models (Lines 164 to 171 of the new version of manuscript).

The other dispersion relationships, however, incorporate different rheological behavior, as the respected Reviewer has mentioned. We need to recall that our main aim in this paper was to formulate these relationships, not the pure elastic model. We believe we needed to make it clearer in the manuscript. The respected Editor had suggested to add statements in the new version of manuscript to make what approach is new in the manuscript. We believe that can also be helpful (Line 88 of the new version of manuscript).

As the respected Reviewer has mentioned, it is interesting to compare the results of recent tests against those of an old dispersion relationship (the RP model). This was a part of the reason that we compared the results of all experimental tests against those of pure elastic model (RP model with an additional added mass term). The other part of the reason was that we found it interesting to discuss the fluid damping coefficients giving the best fitting when different dispersion relationships (pure elastic versus viscolestic) are used. We have clarified it in the new version of manuscript (Lines 164 to 171 of the new version of manuscript).

In relation to justification, we completely agree with the respected Reviewer. Introducing the added mass and fluid damping through radiation problem may make the understanding of these two forces very complicated and dubious, and we may end up in a loop to justify their presence. To avoid this, we have removed the justification and tried to introduce the damping term in the way other scholars did, i.e., we only introduce it as a damping force coefficient (please see line 124 of new version of manuscript). The phrase heave damping is changed to fluid damping in the whole manuscript and the term radiation problem is removed from the whole sections except introduction where we mention finite length problem.

We are also very thankful to the respected Reviewer for noticing the editing error in the solid beam equations (the z*4 term). It is now corrected. We have also replaced z with $\xi$ as it would be easier to follow equations in this case.

In relation to consideration of a continuum model, we agree with the respected Reviewer. If we want to introduce the problem through considering the added mass and damping of radiation problem, then we may unintentionally deliver a message that different scatters have been used to treat the problem (similar to Williams et al. 2017), though we have not used such an approach It is clarified in the manuscript (Line 106 of the new version of manuscript). A fluid damping (Robinson-Palmer combined) with the dissipation inside the ice itself are the only mechanisms

used to calculate the decay rate and no scatterer is assumed in the present research. We have found the research done by Williams et al. (2017) very interesting and have introduced it in the Section 1.

Results
* * *
- right hand columns of fig 2 not needed

We agree with the respected Reviewer. The right-hand column is not needed and is removed (please see the new version of Figure 2).

- why not just have $k_i$ instead of $\alpha$ since the attenuation is only coming from the dispersion relation?

The reason is that authors are also introducing wavenumber in the ice covered sea and use the $k_i$ to represent it. Note that, wavenumber in open-water is used for normalizing decay rate and thus we have used the subscripts $o$ and $i$ to denote the wavenumber in open and ice covered regions, respectively.

- ice rheologies give different attenuation behaviours (peaks in attenuation) at high frequencies. This is interesting that peaks can be produced with different rheologies.

Thanks for this comment and noticing this point. The peak predicted in the decay rate curves is because the decay rate is normalized using open-water wavenumber. The dimensional decay rate only peaks when the RP model is used. Two dimensional figures are presented, and related discussions are added:

Figure 4 and the discussion presented in Lines 372 to 286.

Figure A1 and the discussion presented in Appendix A.

However, once you start to introduce more complexity (I am thinking especially of the SLS models) there are more parameters to be tuned and there is a danger of overfitting.

We agree with the respected Reviewer. Adding more parameters, while interesting, may lead to overfitting of data. Understanding of the mechanical behavior of the ice becomes very hard under such an assumption. Perhaps, more studies need to be carried out in the future to investigate whether a three-parameter model (or even models with greater number of elements) can be used

or not. It is explained in the manuscript. We have made it clear in two different parts of the paper (Lines 450 to 453 and Line 524 to 527 of the new version of manuscript).

Moreover, the peak in attenuation may not be real as instrument noise and local non-linear wave generation of high-frequency waves can give the appearance that high frequencies are being attenuated more than they are (Thompson et al, 2021, J. Geophys. Res.), so trying to fit them too accurately may not be wise.

Thanks for this very constructive comment. Authors believe this point should be explained in the manuscript. As we explained in response to the other comment (the third comment of the respected Reviewer on the Results), only the RP model gives a peak in decay rate plots, though the viscoelastic ones do not. To make it clearer, we have added an Appendix A in which we have presented an example of the dimensional data. It can be seen that the curves constructed using the Maxwell and KV models do not peak in the short-wave regime, though the field data and RP models reach a peak in this zone. Generally, KV and Maxwell models can never predict peak a peak in decay rate plots over the short-wave range.